# Lumina-T2X: Scalable Flow-based Large Diffusion Transformer for Flexible Resolution Generation

**Peng Gao**[1,4][*][‡]  **Le Zhuo**[1,2][*]  **Dongyang Liu**[1,2][*]  **Ruoyi Du**[1][*]  **Xu Luo**[1][*]
**Longtian Qiu**[1]  **Yuhang Zhang**[1]  **Rongjie Huang**[1]  **Shijie Geng**[1]  **Renrui Zhang**[2]
**Junlin Xie**[1]  **Wenqi Shao**[1]  **Zhengkai Jiang**[1]  **Tianshuo Yang**[1]  **Weicai Ye**[1]
**Tong He**[1]  **Jingwen He**[1,2]  **Junjun He**[1]  **Yu Qiao**[1]  **Hongsheng Li**[2,3][†]
[1]Shanghai AI Laboratory   [2]CUHK MMLab   [3]CPII under InnoHK
[4]Shenzhen Institute of Advanced Technology, Chinese Academy of Science

## Abstract

Sora unveils the potential of scaling Diffusion Transformer (DiT) for generating photorealistic images and videos at arbitrary resolutions, aspect ratios, and durations, yet it still lacks sufficient implementation details. In this paper, we introduce the *Lumina-T2X* family – a series of Flow-based Large Diffusion Transformers (Flag-DiT) equipped with zero-initialized attention, as a simple and scalable generative framework that can be adapted to various modalities, *e.g.*, transforming noise into images, videos, multi-view 3D objects, or audio clips conditioned on text instructions. By tokenizing the latent spatial-temporal space and incorporating learnable placeholders such as [nextline] and [nextframe] tokens, Lumina-T2X seamlessly unifies the representations of different modalities across various spatial-temporal resolutions. Advanced techniques like RoPE, KQ-Norm, and flow matching enhance the stability, flexibility, and scalability of Flag-DiT, enabling models of Lumina-T2X to scale up to 7 billion parameters and extend the context window to 128K tokens. This is particularly beneficial for creating ultra-high-definition images with our Lumina-T2I model and long 720p videos with our Lumina-T2V model. Remarkably, Lumina-T2I, powered by a 5-billion-parameter Flag-DiT, requires only 35% of the training computational costs of a 600-million-parameter naive DiT (PixArt-$\alpha$), indicating that increasing the number of parameters significantly accelerates convergence of generative models without compromising visual quality. Our further comprehensive analysis underscores Lumina-T2X's preliminary capability in resolution extrapolation, high-resolution editing, generating consistent 3D views, and synthesizing videos with seamless transitions. All code and checkpoints of Lumina-T2X are released at GitHub to further foster creativity, transparency, and diversity in the generative AI community.

## 1 Introduction

Recent advancements in foundational diffusion models, such as Sora (OpenAI, 2024), Stable Diffusion 3 (Esser et al., 2024), PixArt-$\alpha$ (Chen et al., 2023b), and PixArt-$\Sigma$ (Chen et al., 2024b), have yielded remarkable success in generating photorealistic images and videos. These models demonstrate a paradigm shift from the classic U-Net architecture (Ho et al., 2020) to a transformer-based architecture (Peebles & Xie, 2023a) for diffusion backbones. Notably, with this improved architecture, Sora and Stable Diffusion 3 can generate samples at arbitrary resolutions and exhibit strong adherence to scaling laws, achieving significantly better results with increased parameter sizes. However, they only provide limited guidance on the design choices of their models and lack detailed implementation instructions and publicly available pre-trained checkpoints, limiting their

---

[*]Equal Contribution
[†]Corresponding Authors
[‡]Project Lead

utility for community usage and replication. Moreover, these methods are tailored to specific tasks such as image or video generation tasks, and are formulated from varying perspectives, which hinders potential cross-modality adaptation.

To bridge these gaps, we present **Lumina-T2X**, a family of Flow-based Large Diffusion Transformers (Flag-DiT) capable of efficient and scalable training The largest model within the Lumina-T2X family comprises a Flag-DiT with 7 billion parameters and a multi-modal large language model, SPHINX (Gao et al., 2024; Lin et al., 2023), as the text encoder, with 13 billion parameters, capable of handling 128K tokens. Specifically, the foundational text-to-image model, Lumina-T2I, utilizes the flow matching framework (Liu et al., 2022b; Lipman et al., 2022; Albergo & Vanden-Eijnden, 2022) and is trained on a meticulously curated dataset of high-resolution photorealistic image-text pairs, achieving remarkably realistic results with merely a small proportion of computational resources. As shown in Figure 7, Lumina-T2I can generate high-quality images at arbitrary resolutions and aspect ratios, and further enables advanced functionalities including resolution extrapolation (Du et al., 2024; He et al., 2023), high-resolution editing (Hertz et al., 2022; Brooks et al., 2023; Kawar et al., 2023; Sheynin et al., 2023), compositional generation (Bar-Tal et al., 2023; Yang et al., 2024), and style-consistent generation (Hertz et al., 2023; Tewel et al., 2024), all of which are seamlessly integrated into the framework in a training-free manner. In addition, to empower the generation capabilities across various modalities, Lumina-T2X is independently trained from scratch on video-text, multi-view-text, and speech-text pairs to synthesize videos, multi-view images of 3D objects, and speech from text instructions. The core contributions of Lumina-T2X are summarized as follows:

**Flow-based Large Diffusion Transformers (Flag-DiT)** Lumina-T2X utilizes the Flag-DiT architecture inspired by the core design principles from Large Language Models (LLMs) (Touvron et al., 2023a;b; Brown et al., 2020; Radford et al., 2019; Reid et al., 2024; Team et al., 2023), such as scalable architecture (Brown et al., 2020; Vaswani et al., 2017; Henry et al., 2020; Zhang & Sennrich, 2019; Su et al., 2024; Dehghani et al., 2023) and context window extension (Peng et al., 2023; Su et al., 2024; Chen et al., 2023d; loc, 2024) for increasing parameter size and sequence length. The modifications, including RoPE (Su et al., 2024), RMSNorm (Zhang & Sennrich, 2019), and KQ-Norm (Henry et al., 2020), over the original DiT, significantly enhance the training stability and model scalability, supporting up to 7 billion parameters and sequences of 128K tokens. Moreover, Flag-DiT improves upon the original DiT by adopting the flow matching formulation (Ma et al., 2024; Lipman et al., 2022), which builds continuous-time diffusion paths via linear interpolation between noise and data. We have thoroughly ablated these architecture improvements over the label-conditioned generation on ImageNet (Deng et al., 2009), demonstrating faster training convergence, stable training dynamics, and a simplified training and inference pipeline.

**Versatile Applications within One Framework** By incorporating learnable placeholders such as [nextline] and [nextframe] tokens with RoPE, Lumina-T2X can seamlessly encode any input - regardless of resolution, aspect ratio, or even temporal duration - into a unified 1D token sequence. This design choice unlocks various potential applications by explicitly manipulating the positions of identifiers and position indexes of RoPE during both training and inference. For instance, this flexibility allows for training-free resolution extrapolation, enabling the generation of resolutions surpassing those encountered during training. Lumina-T2I trained at a resolution of $1024 \times 1024$ pixels can generate images ranging from $768 \times 768$ to $1792 \times 1792$ pixels. During training, our 1D sequence modeling framework can adapt to different modalities with minimal modification, avoiding the need for modality-specific architecture design, akin to unified multimodal autoregressive modeling approaches (Lu et al., 2022c; 2024; Team, 2024). We demonstrate how Lumina-T2X supports various training-free text-to-image applications and preliminary exploration of video, multiview, and audio generation in the Appendix.

**Low Training Resources** Our empirical observations indicate that employing larger models, high-resolution images, and longer-duration video clips can significantly accelerate the convergence speed of diffusion transformers. Although increasing the token length prolongs the time of each iteration due to the quadratic complexity of transformers, it substantially reduces the overall training time before convergence by lowering the required number of iterations. Moreover, by utilizing meticulously curated text-image and text-video pairs featuring high aesthetic quality frames and detailed captions (Betker et al., 2023; Chen et al., 2023b; 2024b), our Lumina-T2X model is able to generate high-resolution images and coherent videos with minimal computational demands. It is

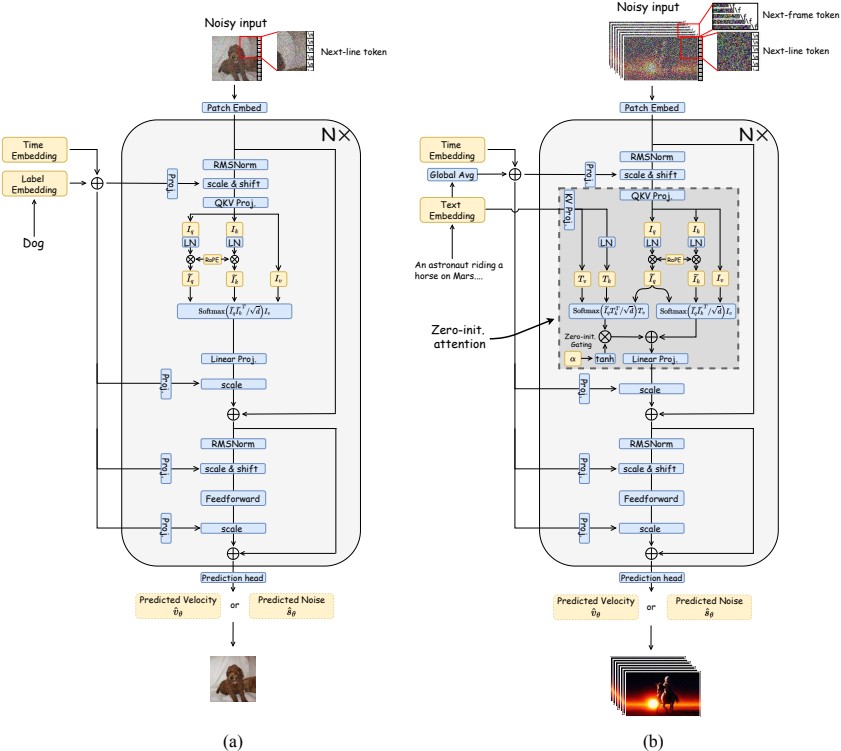

Figure 1: A comparison of Flag-DiT with label and text conditioning. (a) Flag-DiT with label conditioning. (b) Text conditioning with a zero-initialized attention mechanism.

worth noting that the default Lumina-T2I configuration, equipped with a 5 billion Flag-DiT and a 7 billion LLaMA (Touvron et al., 2023a;b) as its text encoder, requires only 35% of the computational resources compared to PixArt-$\alpha$, which builds upon a 600 million DiT backbone and 3 billion T5 (Raffel et al., 2020) as its text encoder. A detailed comparison of computational resources between the default Lumina-T2I and PixArt-$\alpha$ is provided in Table 4.

In this paper, we first introduce the architecture of Flag-DiT and the overall pipeline. We then showcase the results from models in the Lumina-T2X family, accompanied by in-depth analyses. We highly recommend reading the Appendix, where we discuss the Lumina-T2X system in detail, including preliminaries, inference-time applications, and explorations on various modalities. To support future research in the generative AI community, all training, inference codes, and pre-trained models of Lumina-T2X will be released.

## 2 METHOD

### 2.1 FLOW-BASED LARGE DIFFUSION TRANSFORMERS (FLAG-DIT)

DiT (Peebles & Xie, 2023b) is rising to be a popular generative modeling approach with great scaling potential. It operates over latent patches extracted from a pretrained VAE (Kingma & Welling, 2013; Blattmann et al., 2023a), then utilizes a transformer as denoising backbone to predict the mean and variance according to DDPM formulation (Ho et al., 2020; Nichol & Dhariwal, 2021) from different levels of noised latent patches conditioned on time steps and class labels. However, the largest parameter size of DiT is only limited at 600M which is far less than LLMs (e.g., PaLM-540B (Chowdhery et al., 2023; Anil et al., 2023) and LLaMA3-400B (Touvron et al., 2023b)). Besides, DiT requires full precision training which doubles the GPU memory costs and training speed compared with mixed precision training (Micikevicius et al., 2017). Last, the design choice of DiT lacks the flexibility to generate an arbitrary number of images (i.e., videos or multiview images).

To remedy the mentioned problems of DiT, Flag-DiT keeps the overall framework of DiT unchanged while introducing the following modifications to improve scalability, stability, and flexibility.

① **Stability** It is difficult to directly scale up parameter size and token length of DiT due to the instabilities arising in the intermediate stage of training. Flag-DiT builds on top of DiT and incorporates modifications from ViT-22B (Dehghani et al., 2023) and LLaMA (Touvron et al., 2023a;b) to improve the training stability. Specifically, Flag-DiT substitutes all LayerNorm (Ba et al., 2016) with RMSNorm (Zhang & Sennrich, 2019) to improve training stability. Moreover, it incorporates key-query normalization (KQ-Norm) (Dehghani et al., 2023; Henry et al., 2020; Lu et al., 2023) before key-query dot product attention computation. The introduction of KQ-Norm aims to prevent loss divergence by eliminating extremely large values within attention logits (Dehghani et al., 2023). Such simple modifications can prevent divergent loss under mixed-precision training and facilitate optimization with a substantially higher learning rate. The detailed computational flow of Flag-DiT is shown in Figure 1.

② **Flexibility** DiT only supports fixed resolution generation of a single image with simple label conditions and fixed DDPM formulation. To tackle these issues, we first examine why DiT lacks the flexibility to generate samples at arbitrary resolutions and scales. We find this limitation stems from DiT's use of absolute positional embedding (APE) (Dosovitskiy et al., 2020; Touvron et al., 2021), which is added to latent tokens in the first layer, following vision transformers. However, APE, designed for vision recognition tasks, struggles to generalize to unseen resolutions and scales beyond training. Motivated by recent LLMs exhibiting strong context extrapolation capabilities (Peng et al., 2023; Su et al., 2024; Chen et al., 2023d; loc, 2024), we replace APE with RoPE (Su et al., 2024) which injects relative position information in a layerwise manner.

However, the 1D RoPE is insufficient to accurately describe the position of image and video tokens. Therefore, we further introduce learnable special tokens including the `[nextline]` and `[nextframe]` tokens to transform training samples with different scales and durations into a unified one-dimensional sequence. Besides, we add `[pad]` tokens to transform 1D sequences into the same length for better parallelism. This is the key modifications that can significantly improve training and inference flexibility with the support of training or generating samples with arbitrary modality, resolution, aspect ratios, and durations, leading to the final design of Lumina-T2X.

Next, we switch from the DDPM setting in DiT to the flow matching formulation (Ma et al., 2024; Liu et al., 2022b; Lipman et al., 2022), offering another flexibility to Flag-DiT. It is well known the schedule defining how to corrupt data to noise has great impacts on both the training and sampling of standard diffusion models. Thus plenty of diffusion schedules are carefully designed and used, including VE (Song et al., 2020b), VP (Ho et al., 2020), and EDM (Karras et al., 2022). More specifically, given the data $x \sim p(x)$ and Gaussian noise $\epsilon \sim \mathcal{N}(0, I)$, we define an interpolation-based forward process

$$x_t = \alpha_t x + \beta_t \epsilon, \tag{1}$$

where $\alpha_0 = 0$, $\beta_t = 1$, $\alpha_1 = 1$, and $\beta_1 = 0$ to satisfy this interpolation on $t \in [0, 1]$ is defined between $x_0 = \epsilon$ and $x_1 = x$. Similar to the diffusion schedule, this interpolation schedule also offers a flexible choice of $\alpha_t$ and $\beta_t$. For example, we can incorporate the original diffusion schedules, such as $\alpha_t = \sin(\frac{\pi}{2}t), \beta_t = \cos(\frac{\pi}{2}t)$ for VP cosine schedule. In our framework, we adopt the linear interpolation schedule between noise and data for its simplicity, i.e.,

$$x_t = tx + (1 - t)\epsilon. \tag{2}$$

This formulation indicates a uniform transformation with constant velocity between data and noise. The corresponding time-dependent velocity field is given by

$$v_t(x_t) = \dot{\alpha}_t x + \dot{\beta}_t \epsilon \tag{3}$$
$$= x - \epsilon, \tag{4}$$

where $\dot{\alpha}$ and $\dot{\beta}$ denote time derivative of $\alpha$ and $\beta$. This time-dependent velocity field $v : [0, 1] \times \mathbb{R}^d \to \mathbb{R}^d$ defines an ordinary differential equation named Flow ODE

$$dx = v_t(x_t)dt. \tag{5}$$

We use $\phi_t(x)$ to represent the solution of the Flow ODE with the init condition $\phi_0(x) = x$. By solving this Flow ODE from $t = 0$ to $t = 1$, we can transform noise into data sample using the

approximated velocity fields $v_\theta(x_t, t)$. During training, the flow matching objective directly regresses the target velocity

$$\mathcal{L}_v = \int_0^1 \mathbb{E}[\| v_\theta(x_t, t) - \dot{\alpha}_t x - \dot{\beta}_t \epsilon \|^2] dt, \tag{6}$$

which is named Conditional Flow Matching loss (Lipman et al., 2022), sharing similarity with the noise prediction or score prediction losses in diffusion models.

Finally, beyond simple label conditioning for class-conditioned generation, we extend Flag-DiT to flexibly support arbitrary text instruction with zero-initialized attention (Zhang et al., 2023b; Gao et al., 2023; Zhang et al., 2023a; Bachlechner et al., 2021). As shown in Figure 1(b), Flag-DiT leverages the queries of latent image tokens to aggregate information from keys and values of text embeddings. We propose a zero-initialized gating mechanism to gradually inject conditional information into the token sequences. Given image queries $I_q$, keys $I_k$, and values $I_v$ with text keys $T_k$ and values $T_v$, the final attention output is formulated as

$$A = \text{softmax}\left(\frac{\tilde{I}_q \tilde{I}_k^T}{\sqrt{d}}\right) I_v + \tanh(\alpha)\, \text{softmax}\left(\frac{\tilde{I}_q T_k^T}{\sqrt{d}}\right) T_v, \tag{7}$$

where $\tilde{I}_q$ and $\tilde{I}_k$ stand for applying RoPE defined in Equation 8 to image queries and values, $d$ is the dimension of queries and keys, and $\alpha$ indicates the zero-initialized learnable parameter in gated cross-attention. In our experiments, we discovered that zero-initialized attention induces sparsity gating which can turn off 90% text embedding conditions across layers and heads. This indicates the potential for designing more efficient T2I models in the future.

Equipped with the above improvements, our Flag-DiT supports arbitrary resolution generation of multiple images with arbitrary conditioning using a unified flow matching paradigm.

③ **Scalability** After alleviating the training stability of DiT and increasing flexibility for supporting arbitrary resolutions conditioned on text instructions, we empirically scale up Flag-DiT with larger parameters and more training samples. Specifically, we explore scaling up the parameter size from 600M to 7B on the label-conditioned ImageNet generation benchmark. The detailed configurations of Flag-DiT with different parameter sizes are discussed in Appendix D. Flag-DiT can be stably trained under mixed-precision configuration and achieve fast convergence compared with vanilla DiT as shown in the experiment section. After verifying the scalability of our Flag-DiT model, we scale up the token length to 4K and expand the dataset from label-conditioned 1M ImageNet to more challenging 14M high-resolution image-text pairs. We further successfully verified that Flag-DiT can support the generation of long videos up to 128 frames, equivalent to 128K tokens. As Flag-DiT is a pure transformer-based architecture, it can borrow the well-validated parallel strategies designed for LLMs, including FSDP (Zhao et al., 2023) and sequence parallel (Liu et al., 2023a; Liu & Abbeel, 2024; Liu et al., 2024; Jacobs et al., 2023) to support large parameter scales and longer sequences. Therefore, we can conclude that Flag-DiT is a scalable generative model with respect to model parameters, sequence length, and dataset size.

## 2.2 THE OVERALL PIPELINE OF LUMINA-T2X

As illustrated in Figure 2, the pipeline of Lumina-T2X consists of four main components during training, which will be described below.

**Frame-wise Encoding of Different Modalities** The key ingredient for unifying different modalities within our framework is treating images, videos, multi-view images, and speech spectrograms as frame sequences of length $T$. We can then utilize modality-specific encoders, to transform these inputs into latent frames of shape $[H, W, T, C]$. Specifically, for images ($T = 1$), videos ($T = \texttt{numframes}$), and multiview images ($T = \texttt{numviews}$), we use SD 1.5 VAE to independently encode each image frame into latent space and concatenate all latent frames together, while we leave speech spectrograms unchanged using identity mapping. Our approach establishes a universal data representation that supports diverse modalities, enabling our Flag-DiT to effectively model.

**Text Encoding with Diverse Text Encoders** For text-conditional generation, we encode the text prompts using pre-trained language models. Specifically, we incorporate a variety of diverse text

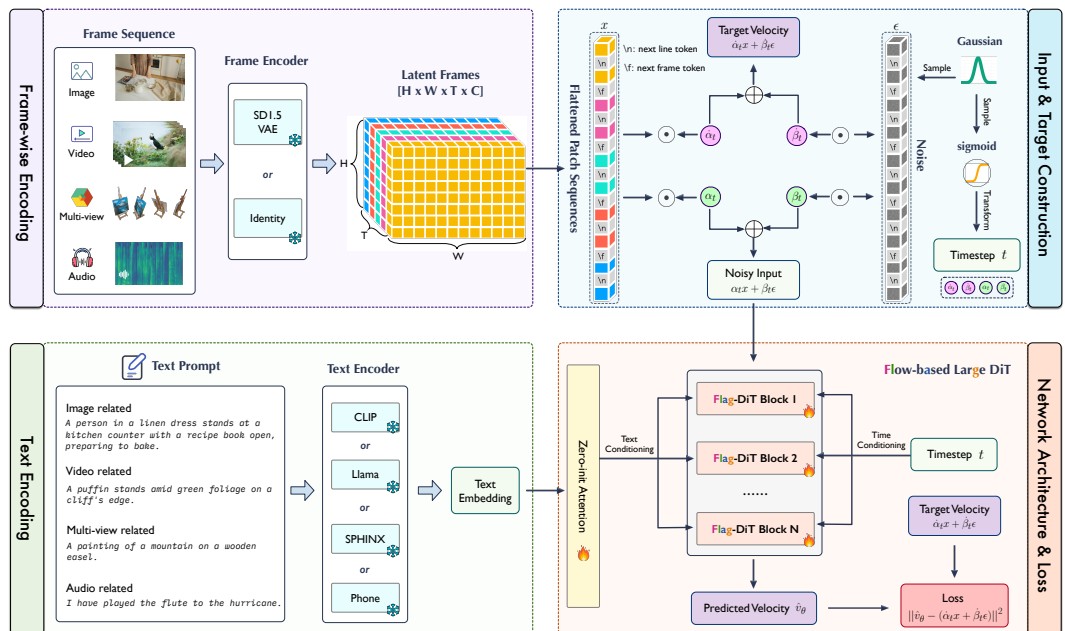

Figure 2: Our Lumina-T2X framework consists of four components: frame-wise encoding, input & target construction, text encoding, and prediction based on Flag-DiT.

encoders with varying sizes, including CLIP, LLaMA, SPHINX, and Phone encoders, tailored for various needs and modalities, to optimize text conditioning. We provided a series of Lumina-T2X trained with different text encoders mentioned above in our model zoo as shown in Figure 6.

**Input & Target Construction** As described in Section 2.1, latent frames are first flattened using $2 \times 2$ patches into a 1D sequence, then added with [nextline] and [nextframe] tokens as identifiers. Lumina-T2X adopts the linear interpolation schedule in flow-matching to construct the input and target following Equations 2 and 4 for its simplicity and flexibility. Inspired by the observation that intermediate timesteps are critical for both diffusion models (Karras et al., 2022) and flow-based models (Esser et al., 2024), we adopt the time resampling strategy to sample timestep from a log-norm distribution during training. Specifically, we first sample a timestep from a normal distribution $\mathcal{N}(0, 1)$ and map it to $[0, 1]$ using the logistic function in order to emphasize the learning of intermediate timesteps.

**Network Architecture & Loss** We use Flag-DiT as our denoising backbone with detailed architecture of each Flag-DiT block depicted in Figure 1. Given the noisy input, the Flag-DiT Blocks inject diffusion timestep added with global text embedding through a modulation mechanism and further integrate text conditioning via zero-initialized attention defined by Equation 7. We apply RMSNorm at the beginning of each attention and MLP block and use KQ-Norm for key and query vectors to prevent uncontrolled growth in absolute values, which can lead to numerical instability. Finally, we compute the regression loss between predicted velocity and ground-truth velocity using the Conditional Flow Matching loss defined in Equation 6.

## 3 EXPERIMENTS

### 3.1 VALIDATING FLAG-DIT ON IMAGENET

**Training Setups** We perform experiments on label-conditioned 256×256 and 512×512 ImageNet (Deng et al., 2009) generation to validate the advantages of Flag-DiT over DiT (Peebles & Xie, 2023b). We train a specialized version of Flag-DiT, *i.e.*, Flag-DiT-D, which adopts the original DDPM formulation (Ho et al., 2020; Nichol & Dhariwal, 2021) in DiT to enable a fair comparison with the original DiT. We exactly follow the setups of DiT but with the following modifications, including, mixed precision training, large learning rate, and architecture modifications suite (*e.g.*

Table 1: Full comparison between Flag-DiT-D and Flag-DiT with other models on ImageNet $256 \times 256$ and $512 \times 512$ label-conditional generation. Notably, -D indicates Flag-DiT using the original diffusion schedule. P, R, and -G denote Precision, Recall, and results with classifier-free guidance, respectively. We also include the total number of images during the training stage to offer further insights into the convergence speed of different generative models.

**ImageNet 256×256 Benchmark**

| Models | #Images (M) | FID ↓ | sFID ↓ | IS ↑ | P ↑ | R ↑ |
|---|---|---|---|---|---|---|
| BigGAN-deep (Brock et al., 2018) | - | 6.95 | 7.36 | 171.40 | 0.87 | 0.28 |
| MaskGIT (Chang et al., 2022) | 355 | 6.18 | - | 182.1 | 0.80 | 0.51 |
| StyleGAN-XL (Sauer et al., 2022) | - | 2.30 | 4.02 | 265.12 | 0.78 | 0.53 |
| ADM (Dhariwal & Nichol, 2021) | 507 | 10.94 | 6.02 | 100.98 | 0.69 | 0.63 |
| ADM-U (Dhariwal & Nichol, 2021) | 507 | 7.49 | **5.13** | 127.49 | **0.72** | 0.63 |
| LDM-8 (Rombach et al., 2022) | 307 | 15.51 | - | 79.03 | 0.65 | 0.63 |
| LDM-4 (Rombach et al., 2022) | 213 | 10.56 | - | 103.49 | 0.71 | 0.62 |
| DiffuSSM-XL (Yan et al., 2023) | 660 | 9.07 | 5.52 | 118.32 | 0.69 | 0.64 |
| DiT-XL/2 (Peebles & Xie, 2023b) | 1792 | 9.62 | 6.85 | 121.50 | 0.67 | 0.67 |
| SiT-XL/2 (Ma et al., 2024) | 1792 | 8.60 | - | - | - | - |
| **Flag-DiT-D-7B** | 256 | **6.09** | 5.59 | **153.32** | 0.70 | **0.68** |
| Classifier-Free Guidance | | | | | | |
| ADM-G (Dhariwal & Nichol, 2021) | 507 | 4.59 | 5.25 | 186.70 | 0.82 | 0.52 |
| ADM-G, ADM-U (Dhariwal & Nichol, 2021) | 507 | 3.60 | - | 247.67 | **0.87** | 0.48 |
| LDM-8-G (Rombach et al., 2022) | 307 | 7.76 | - | 209.52 | 0.84 | 0.35 |
| LDM-4-G (Rombach et al., 2022) | 213 | 3.95 | - | 178.22 | 0.81 | 0.55 |
| U-ViT-H/2-G (Bao et al., 2023) | 512 | 2.29 | - | 247.67 | **0.87** | 0.48 |
| DiT-XL/2-G (Peebles & Xie, 2023b) | 1792 | 2.27 | 4.60 | 278.24 | 0.83 | 0.57 |
| DiffuSSM-XL-G (Yan et al., 2023) | 660 | 2.28 | 4.49 | 259.13 | 0.86 | 0.56 |
| SiT-XL/2-G (Ma et al., 2024) | 1792 | 2.06 | 4.50 | 270.27 | 0.82 | 0.59 |
| **Flag-DiT-D-3B-G** | 435 | 2.10 | 4.52 | **304.36** | 0.82 | 0.60 |
| **Flag-DiT-3B-G** | 256 | **1.96** | **4.43** | 284.80 | 0.82 | **0.61** |
| **ImageNet 512×512 Benchmark** | | | | | | |
| ADM (Dhariwal & Nichol, 2021) | 1385 | 23.24 | 10.19 | 58.06 | 0.73 | 0.60 |
| ADM-U (Dhariwal & Nichol, 2021) | 1385 | 9.96 | 5.62 | 121.78 | 0.75 | **0.64** |
| ADM-G (Dhariwal & Nichol, 2021) | 1385 | 7.72 | 6.57 | 172.71 | **0.87** | 0.42 |
| ADM-G, ADM-U (Dhariwal & Nichol, 2021) | 1385 | 3.85 | 5.86 | 221.72 | 0.84 | 0.53 |
| U-ViT/2-G (Bao et al., 2023) | 512 | 4.05 | 8.44 | 261.13 | 0.84 | 0.48 |
| DiT-XL/2-G (Peebles & Xie, 2023b) | 768 | 3.04 | 5.02 | 240.82 | 0.84 | 0.54 |
| DiffuSSM-XL-G (Yan et al., 2023) | 302 | 3.41 | 5.84 | 255.06 | 0.85 | 0.49 |
| **Flag-DiT-D-3B-G** | 472 | **2.52** | **5.01** | **303.70** | 0.82 | 0.57 |

KQ-Norm, RoPE, and RMSNorm). By default, we report FID-50K (Parmar et al., 2022; Dhariwal & Nichol, 2021) using 250 DDPM sampling steps for Flag-DiT-D and the adaptive Dopri-5 solver for Flag-DiT. We additionally report sFID (Salimans et al., 2016), Inception Score (Nash et al., 2021), and Precision/Recall (Kynkäänniemi et al., 2019) for an extensive evaluation.

**Comparison with SOTA Approaches**   As shown in Table 1, Flag-DiT-D-7B significantly surpasses all approaches on FID and IS score without using classifier-free guidance (CFG) (Ho & Salimans, 2022), reducing the FID score from 8.60 to 6.09. This indicates increasing the parameters of diffusion models can significantly improve the sample quality without relying on extra tricks such as CFG. When CFG is employed, both Flag-DiT-D-3B and Flag-DiT-3B achieve slightly better FID scores but much improved IS scores than DiT-600M and SiT-600M while only requiring 24% and 14% training iterations. For 512×512 label-conditioned ImageNet generation, Flag-DiT-D with 3B parameters significantly surpass other SOTA approaches by reducing FID from 3.04 to 2.52 and increasing IS from 240 to 303. This validates that increased parameter scale can better capture complex high-resolution details. By comparison with SOTA approaches on label-conditioned ImageNet generation, we can conclude that Flag-DiT-D and Flag-DiT are good at generative modeling with fast convergence, stable scalability, and strong high-resolution modeling ability. This directly motivates Lumian-T2X

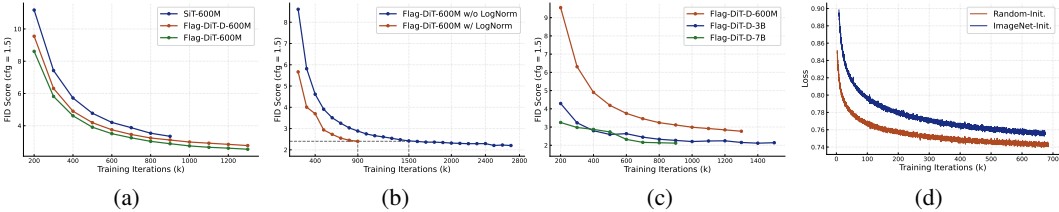

(a)        (b)        (c)        (d)

Figure 3: Training dynamics of different configurations, to explore the effects of (a) flow matching formulation and architecture modifications, (b) using LogNorm sampling, (c) scaling up model size, and (d) using ImageNet initialization.

to employ Flag-DiT with large parameters to model more complex generative tasks for any modality, resolution, and duration generation.

**Comparison between Flag-DiT, Flag-DiT-D, and SiT**    We compared the performance of Flag-DiT, Flag-DiT-D, and SiT on ImageNet-256 benchmark, fixing the parameter size at 600M for a fair comparison. As demonstrated in Figure 3(a), Flag-DiT consistently outperforms Flag-DiT-D across all epochs in FID evaluation. This indicates that the flow matching formulation can improve image generation compared to the standard diffusion setting. Moreover, Flag-DiT's lower FID scores compared to SiT suggest that meta-architecture modifications, including RMSNorm, RoPE, and K-Q norm, not only stabilize training but also boost performance.

**Faster Training Speed with Mixed Precision Training**    Flag-DiT not only improves performance but also enhances training efficiency as well as stability. Unlike DiT, which diverges under mixed precision training, Flag-DiT can be trained stably with mixed precision. Thus Flag-DiT leads to faster training speeds compared with DiT at the same parameter size. We measure the throughputs of 600M and 3B Flag-DiT and DiT on one A100 node with 256 batch size. As shown in Table 3. Flag-DiT can process 40% more images per second.

**Faster Convergence with LogNorm Sampling**    During training, Flag-DiT-600M uniformly samples time steps from 0 to 1. Previous works (Karras et al., 2022; Esser et al., 2024) have pointed out that the learning of score function in diffusion models or velocity field in flow matching is more challenging in the middle of the schedule. To address this, we have replaced uniform sampling with log-normal sampling, which places greater emphasis on the central time steps, thereby accelerating convergence. We refer to the Flag-DiT-600M model using log-normal sampling as Flag-DiT-600M-LogNorm. As demonstrated in Figure 3(b), Flag-DiT-600M-LogNorm not only achieves faster loss convergence but also improves the FID score significantly.

**Scaling Effects of Flag-DiT**    DiT demonstrates that the quality of generated images improves with an increase in parameters. However, the largest DiT model tested is limited to 600M parameters, significantly fewer than those used in large language models. Previous experimental sessions have validated the stability, effectiveness, and rapid convergence of Flag-DiT. Building on this foundation, we have scaled the parameters of Flag-DiT from 600M to 7B while maintaining the same hyperparameters. As depicted in Figure 3(c), this substantial increase in parameters significantly enhances the convergence speed of Flag-DiT, indicating that larger models are more compute-efficient for training.

**Influence of ImageNet Initialization**    PixArt-$\alpha$ (Chen et al., 2023b; 2024b) utilizes ImageNet-pretrained DiT, which learns pixel dependency, as an initialization for the subsequent T2I model. To validate the influence of ImageNet initialization, we compare the velocity prediction loss of Lumina-T2I with a 600M parameter model using ImageNet initialization versus training from scratch. As illustrated in Figure 3(d), training from scratch consistently results in lower loss levels and faster convergence speeds. Moreover, starting from scratch allows for a more flexible choice of configurations and architectures, without the constraints of a pretrained network. This observation also leads to the design of simple and fast training recipes shown in Table 4.

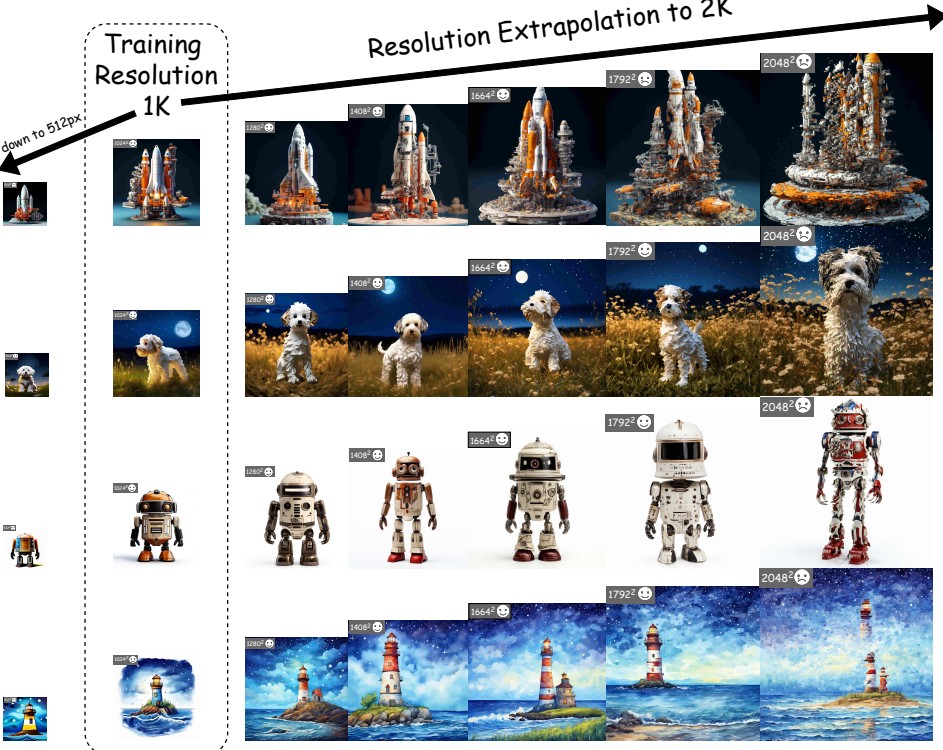

Figure 4: Resolution extrapolation samples of Lumina-T2I. Without any additional training, Lumina-T2I is capable of directly generating images with various resolutions from $512^2$ to $1792^2$.

## 3.2 RESULTS OF LUMINA-T2I

**Basic Setups**    Lumina-T2I is a key component of the Lumina-T2X family. By default, all images in this paper are generated using a 5B Flag-DiT coupled with a 7B LLaMA text encoder (Touvron et al., 2023a;b). The Lumina-T2I model zoo also supports various text encoder sizes, DiT parameters, input and target construction, and latent spaces, as shown in Appendix D. Lumina-T2I models are progressively trained on images with resolutions of 256, 512, and 1024. Detailed information on batch size, learning rate, and computational costs for each stage is provided in Table 4.

**Fundamental Text-to-Image Generation Ability**    We showcase the fundamental text-to-image generation capability in Figure 8 and Figure 7. The large capacity of the diffusion backbone and text encoder allows for the generation of photorealistic, high-resolution images with accurate text comprehension, utilizing just 288 A100 GPU days. By explicitly indicating the placement of [nextline] tokens during inference, Lumina-T2I can flexibly generate images from text instructions of various sizes.

**Tuning-Free Resolution-Extrapolation**    Due to exponential growth in computational demand and data scarcity, existing T2I models are generally limited to 1K resolution. Thus, there is a significant demand for low-cost and high-resolution extrapolation approaches (He et al., 2023; Du et al., 2024; Cheng et al., 2024). The translational invariance of RoPE enhances Lumina-T2I's potential for resolution extrapolation, allowing it to generate images at out-of-domain resolutions. Inspired by the practices in previous arts, we adopt three techniques that can help unleash Lumina-T2I's potential of test-time resolution extrapolation: (1) NTK-aware scaled RoPE (loc, 2024) that rescales the rotary base of RoPE to achieve a gradual position interpolation of the low-frequency components, (2) Time Shifting (Esser et al., 2024) that reschedules the timesteps to ensure consistent SNR across denoising processes of different resolutions, and (3) Proportional Attention (Jin et al., 2024) that rescales the attention score to ensure stable attention entropy across various sequence lengths. The details about the aforementioned techniques in our implementation can be found in Appendix F.1.

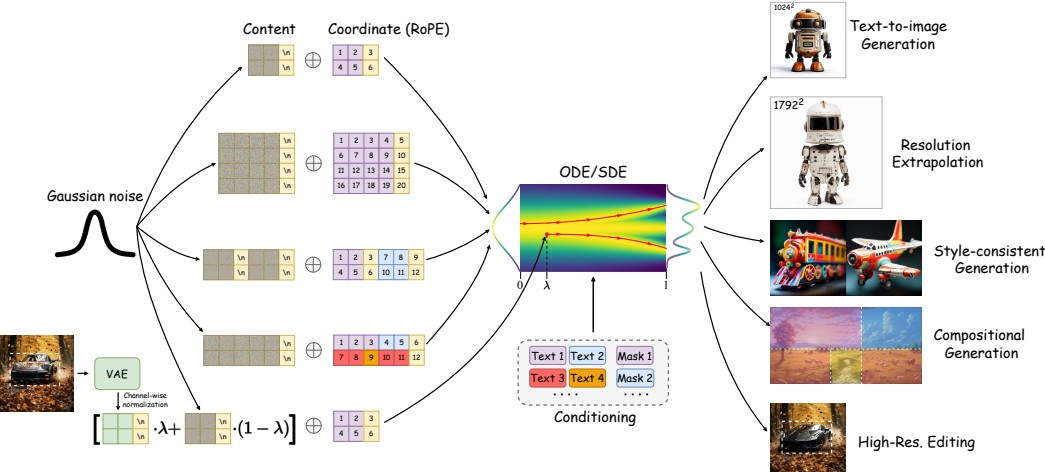

Figure 5: Lumina-T2I supports T2I generation, resolution extrapolation, style-consistent generation, compositional generation, and high-resolution editing in a unified and training-free framework.

Resolution extrapolation brings not only larger-scale images but also higher image quality along with enhanced details. As shown in Figure 4, we observe the quality of generated images and text-to-image alignments can be significantly enhanced as we perform resolution extrapolation from 1K to 1.5K. Besides, Lumina-T2I is also capable of performing extrapolation to generate images with lower resolutions, such as 512 resolution, offering additional flexibility. Conversely, PixArt-$\alpha$ (Chen et al., 2023b), which uses standard positional embeddings instead of RoPE (Su et al., 2024), does not show comparable generalization capabilities at test resolutions. Further enhancing the resolution from 1.5K to 2K can gradually lead to the failure of image generation due to the large domain gap between training and inference. The improvement of image quality and text-to-image alignment is a free lunch of Lumina-T2I as it can improve image generation without incurring any training costs. However, as expected, the free lunch is not without its shortcomings. The discrepancy between the training and inference domains can introduce minor artifacts. We believe the artifacts can be alleviated by collecting high-quality images larger than 1K resolution and performing few-shot parameter-efficient fine-tuning.

## 3.3 MORE ADVANCED APPLICATIONS OF LUMINA-T2X

Beyond its basic text-to-image generation capabilities and resolution extrapolation, Lumina-T2X supports more complex content creations in various modalities as a foundational model. As shown in Figure 5, our pre-trained Lumina-T2X can perform advanced visual tasks including style-consistent generation, high-resolution image editing, and compositional generation – all in a tuning-free manner. Unlike previous methods that address these tasks with varied approaches, Lumina-T2X uniformly tackles these problems through token operations, as depicted in Appendix F. Additionally, we also provide details and results of Lumina-T2V, Lumina-T2MV, and Lumina-T2Speech in Appendix H.

## 4 CONCLUSION

In this paper, we present Lumina-T2X, a unified framework designed for scalable and efficient generation. At the core of Lumina-T2X is a series of Flow-based Large Diffusion Transformers (Flag-DiT) carefully designed for scalable conditional generation. Equipped with key modifications including RoPE, RNSNorm, KQ-Norm, and zero-initialized attention for model architecture, `[nextline]` and `[nextframe]` tokens for data representation, and switching from diffusion to flow matching formulation, our Flag-DiT showcases great improvements in stability, flexibility, and scalability compared to the origin diffusion transformer. We demonstrate the fondational generation capability of Lumina-T2X on the ImageNet benchmark as well as text-to-image, video, multiview, and speech generation. Overall, we hope that our attempts, findings, and open-sources of Lumina-T2X can help clarify the roadmap of generative AI and serve as a new starting point for further research into developing effective large-scale multi-modal generative models.

## 5 ACKNOWLEDGMENT

This project is funded in part by the National Natural Science Foundation of China (No. 62206272), by the National Key R&D Program of China Project 2022ZD0161100, by the Centre for Perceptual and Interactive Intelligence (CPII) Ltd under the Innovation and Technology Commission (ITC)'s InnoHK, by NSFC-RGC Project N_CUHK498/24. Hongsheng Li is a PI of CPII under the InnoHK.

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

## A LIMITATIONS AND FUTURE WORK

**Unified Framework but Independent Training**  Unlike autoregressive models (Lu et al., 2024), which utilize a unified discrete token representation, the latent feature distributions differ significantly across modalities. For example, while joint training with images and videos can enhance visual quality, it may negatively affect the dynamic aspects of video generation. Furthermore, multiview and audio representations vary even more significantly. The disparity in data quantity—where high-quality image data is more abundant than video data, which in turn exceeds multiview data—poses additional challenges for joint training. Therefore, the current version of Lumina-T2X is separately trained to tackle the generation of images, videos, multi-views of 3D objects, and speech. Without leveraging the pre-trained weights on 2D images, Lumina-T2V and Lumina-T2MV achieve preliminary results on temporal- or view-consistent generation but show inferior sample qualities compared with their counterparts. Currently, we propose Lumina-T2X as a unified framework for scaling up models across any modality. In the future, we will further explore the joint training of images, videos, multi-views and audio for better generation quality and fast convergence.

**Fast Convergence but Inadequate Data Coverage**  Although the large model size enables Lumina-T2X to achieve generative capabilities comparable to its counterparts with fast convergence, there remains a limitation in the inadequate coverage of the diverse data spectrum by the collected data. This leads to incomplete learning of the complex patterns and nuances of the real physical world, which can result in less robust model performance in real-world scenarios. Therefore, Lumina-T2X also faces common issues of current generative models, such as struggling with generating detailed human structures like hands or encountering artificial noises and background blurring in complex scenes, leading to less realistic images. We believe that higher-quality real-world data, combined with Lumina-T2X's powerful convergence capabilities, will be an effective solution to address this issue.

## B RELATED WORK

**AI-Generated Contents (AIGCs)**  Generating high-dimensional perceptual data content (*e.g.*, images, videos, audio, *etc*) has long been a challenge in the field of artificial intelligence. In the era of deep learning, Generative Adversarial Networks (GANs) (Goodfellow et al., 2014; Zhu et al., 2017; Isola et al., 2017; Wang et al., 2018; Brock et al., 2018; Karras et al., 2019) stand as a pioneering method in this field due to their efficient sampling capabilities, yet they face issues of training instability and mode collapse. Meanwhile, Variational Autoencoders (VAEs) (Kingma & Welling, 2013; Kusner et al., 2017; An & Cho, 2015; Vahdat & Kautz, 2020; Shao et al., 2020) and flow-based models (Dinh et al., 2014; 2016) demonstrate better training stability and interpretability but lag behind GANs in terms of image quality. Following this, autoregressive models (ARMs) (Van Den Oord et al., 2016; Van den Oord et al., 2016; Child et al., 2019; Chen et al., 2020a) have shown exceptional performance but come with higher computational demands, and the sequential sampling mechanism is more suited to 1D data.

Nowadays, Diffusion Models (DMs) (Sohl-Dickstein et al., 2015), learning to invert diffusion paths from real data towards random noise, have gradually become the de-facto approach of generative AI across multiple domains, with numerous practical applications (OpenAI, a; Anthropic; Google; OpenAI, b; mid; Podell et al., 2023; Esser et al., 2024; run). The success of diffusion models over the past four years can be attributed to the progress in several areas, including reformulating diffusion models to predict noise instead of pixels (Ho et al., 2020), improvements in sampling methods for better efficiency (Song et al., 2020a; Lu et al., 2022a;b; Karras et al., 2022), the introduction of classifier-free guidance that enables direct conversion of text to images (Ho & Salimans, 2021), and cascaded/latent space models that reduce the computational cost of high-resolution generation (Ho et al., 2022b; Rombach et al., 2022; Teng et al., 2023). Apart from generating high-quality images following text instruction, various applications, including high-resolution generation(He et al., 2023; Du et al., 2024; Hwang et al., 2024; Zheng et al., 2024a; Cheng et al., 2024; Chen et al., 2024b), compositional generation (Jiménez, 2023; Bar-Tal et al., 2023; Yang et al., 2024), style-consistent generation (Hertz et al., 2023; Tewel et al., 2024), image editing (Hertz et al., 2022; Brooks et al., 2023; Kawar et al., 2023; Mokady et al., 2023), and controllable generation (Zhang et al., 2023a; Mou et al., 2024; Zhao et al., 2024; Mo et al., 2023), have been proposed to further extend the applicability of pretrained T2I models. Additionally, pre-trained T2I models are also applied with a decoupled

temporal attention to generate videos (Guo et al., 2023; Wang et al., 2023c; Jiang et al., 2023; Chen et al., 2023a; 2024a; Blattmann et al., 2023c; Zhou et al., 2023; Ho et al., 2022a; Blattmann et al., 2023b; Hu et al., 2023; Chen et al., 2023c; Wang et al., 2023b; Zhang et al., 2023c; Xing et al., 2023; Gupta et al., 2023; Wu et al., 2023) and multi-views of 3D object (Shi et al., 2023b; Li et al., 2023; Wang & Shi, 2023; Zuo et al., 2024; Chen et al., 2024d; Voleti et al., 2024; Han et al., 2024; Long et al., 2023; Tang et al., 2024; Liu et al., 2023c; Shi et al., 2023a). The similar framework, with suitable adjustments, has also been applied to audio generation (Huang et al., 2023a; Liu et al., 2023b; Ghosal et al., 2023; Yang et al., 2023). Although this paradigm has achieved notable success at the current model scale (Podell et al., 2023; Pernias et al., 2023; Zheng et al., 2024b), subsequent works have proven the better potential of diffusion models based on vision transformers (so-called Diffusion Transformer, DiT) (Peebles & Xie, 2023b). Afterwards, SiT (Ma et al., 2024) and SD3 (Esser et al., 2024) further demonstrate that an interpolation or flow-matching framework (Liu et al., 2022b; Lipman et al., 2022; Albergo & Vanden-Eijnden, 2022; Albergo et al., 2023) can better enhance the stability and scalability of DiT — pointing the way for diffusion models to scale up to the next level.

Very recently, Sora (OpenAI, 2024) has demonstrated the potential for scaling DiT with its powerful joint image and video generation capabilities. However, the detailed implementations have yet to be released. Therefore, inspired by Sora, we introduce Lumina-T2X to push the boundaries of open-source generative models by scaling the flow-based Diffusion Transformer to generate contents across any modalities, resolutions, and durations.

## C  PRELIMINARIES OF LUMINA-T2X

In this section, we revisit several milestone studies on leveraging diffusion transformers for text-to-image and text-to-video generation, as well as seminal research on large language models (LLMs), all of which greatly inspired the design of Lumina-T2X.

**Rotary Position Embedding (RoPE)**   RoPE (Su et al., 2024) is a type of position embedding that can encode relative positions within self-attention operations. It can be regarded as a multiplicative bias based on position – given a sequence of the query/key vectors, the $n$-th query and the $m$-th key after RoPE can be expressed as:

$$\tilde{q}_m = f(q_m, m) = q_m e^{im\Theta}, \quad \tilde{k}_n = f(k_n, n) = k_n e^{in\Theta}, \tag{8}$$

where $\Theta$ is the frequency matrix. Equipping with RoPE, the calculation of attention scores can be considered as taking the real part of the standard Hermitian inner product:

$$\text{Re}[f(q_m, m)f^*(k_n, n)] = \text{Re}[q_m k_n^* e^{i\Theta(m-n)}]. \tag{9}$$

In this way, the relative position $m - n$ between the $m$-th and $n$-th tokens can be explicitly encoded. Compared to absolute positional encoding, RoPE offers translational invariance, which can enhance the context window extrapolation potential of LLMs. Many subsequent techniques further explore and unlock this potential, *e.g.*, position interpolation (Chen et al., 2023d), NTK-aware scaled RoPE (loc, 2024), Yarn (Peng et al., 2023), *etc*. In this work, Flag-DiT applies RoPE to the keys and queries of diffusion transformers. Notably, this simple technique endows Lumina-T2X with superior resolution extrapolation potential (*i.e.*, generating images at out-of-domain resolutions unseen during training), as demonstrated in Section 3.2, compared to its competitors.

**DiT, Scalable Interpolant Transformer (SiT) and Flow Matching**   U-Net has been the de-facto diffusion backbone in previous Denoising Diffusion Probabilistic Models (Ho et al., 2020) (DDPM). DiT (Peebles & Xie, 2023a) explores using transformers trained on latent patches as an alternative to U-Net, achieving state-of-the-art FID scores on class-conditional ImageNet benchmarks and demonstrating superior scaling potentials in terms of training and inference FLOPs. Furthermore, SiT (Ma et al., 2024) utilizes the stochastic interpolant framework (Albergo et al., 2023) (or flow matching (Lipman et al., 2022)) to connect different distributions in a more flexible manner than DDPM. Extensive ablation studies by SiT reveal that linearly connecting two distributions, predicting velocity fields, and employing a stochastic solver can enhance sample quality with the same DiT architecture. However, both DiT and SiT are limited in model sizes, up to 600 million parameters, and suffer from training instability when scaling up. Therefore, we borrow design choices from LLMs and validate that simple modifications can train a 7-billion-parameter diffusion transformer in mixed precision training.

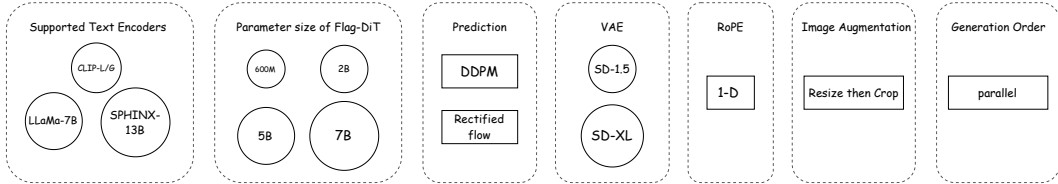

Figure 6: Configurations of Lumina-T2X, including choices of text encoders, parameter sizes for Flag-DiT, prediction targets, VAEs of various sizes, RoPE, image augmentation policies, and generation orders.

Table 2: Detailed configurations of our Flag-DiT backbone.

| Model | Hidden Size | Heads | Layers |
|---|---|---|---|
| Flag-DiT-S | 4 | 8 | 768 |
| Flag-DiT-B | 8 | 12 | 768 |
| Flag-DiT-L | 12 | 24 | 1024 |
| Flag-DiT-XL | 20 | 28 | 1152 |
| Flag-DiT-5B | 32 | 32 | 3072 |
| Flag-DiT-7B | 32 | 32 | 4096 |

Table 3: Training throughput as measured with ImageNet on a single $8 \times$ A100 machine.

| Model | Resolution | Throughput (imgs/s) |
|---|---|---|
| DiT-XL | 256 | 435 |
| DiT-XL | 512 | 80 |
| DiT-XL | 1024 | 10 |
| Flag-DiT-XL | 256 | 600 |
| Flag-DiT-5B | 256 | 195 |
| Flag-DiT-5B | 512 | 32 |
| Flag-DiT-5B | 1024 | 9 |
| Flag-DiT-7B | 256 | 120 |

**PixArt-$\alpha$ and -$\Sigma$** DiT explores the potential of transformers for label-conditioned generation. Built on DiT, PixArt-$\alpha$ (Chen et al., 2023b) unleashes this potential for generating images based on arbitrary textual instructions. PixArt-$\alpha$ significantly reduces training costs compared with SDXL (Podell et al., 2023) and Raphael (Xue et al., 2024), while maintaining high sample quality. This is achieved through multi-stage progressive training, efficient text-to-image conditioning with DiT, and the use of carefully curated high-aesthetic datasets. PixArt-$\Sigma$ extends this approach by increasing the image generation resolution to 4K, facilitated by the collection of 4K training image-text pairs.

Lumina-T2I is highly motivated by PixArt-$\alpha$ and -$\Sigma$ yet it incorporates several key differences. Firstly, Lumina-T2I utilizes Flag-DiT with 5B parameters as the backbone, which is 8.3 times larger than the 0.6B-parameter backbone used by PixArt-$\alpha$ and -$\Sigma$. According to studies on class-conditional ImageNet generation in Section 3.1, larger diffusion models tend to converge much faster than their smaller counterparts and excel at capturing details on high-resolution images. Secondly, unlike PixArt-$\alpha$ and -$\Sigma$ that were pretrained on ImageNet (Deng et al., 2009) and SAM-HD (Kirillov et al., 2023) images, Lumina-T2I is trained directly on high-aesthetic synthetic datasets without being interfered by the domain gap between images from different domains. Thirdly, while PixArt-$\alpha$ and -$\Sigma$ excel at generating images with the same resolution as training stages, our Lumina-T2I, through the introduction of RoPE and [nextline] token, possesses a resolution extrapolation capability, enabling generating images at a lower or higher resolution unseen during training, which offers a significant advantage in generating and transferring images across various scales.

**Sora** Sora (OpenAI, 2024) demonstrates remarkable improvements in text-to-video generation that can create 1-minute videos with realistic or imaginative scenes spanning different durations, resolutions, and aspect ratios. In comparison, Lumina-T2V can also generate 720p videos at arbitrary aspect ratios. Although there still exists a noticeable gap in terms of video length and quality between Lumian-T2V and Sora, video samples from Lumina-T2V exhibit considerable improvements over open-source models on scene transitions and alignment with complex text instructions. We have released all codes of Lumina-T2V and believe training with more computational resources, carefully designed spatial-temporal video encoder, and meticulously curated video-text pairs will further elevate the video quality.

## D    DIVERSE CONFIGURATIONS

The Lumina-T2X family supports a diverse range of configurations, as depicted in Figure 6. Each configuration is independently trained, following the setups outlined in the main text.  For the denoising backbone, we provide multiple Flag-DiT configurations that span a wide range of model sizes from 600M to 7B to provide a trade-off between inference speed and quality, detailed in Table 2. Table 3 demonstrates that our Flag-DiT achieves around 50% faster throughput than the original DiT with the same model size. Notably, Flag-DiT-5B attains throughput speeds comparable to the DiT-XL at the resolution of 1024, showcasing its efficiency in dealing with high-resolution image generation. Regarding the text encoder, we include options such as CLIP-L/G, LLaMA-7B, and SPHINX-13B, which balance GPU consumption with advanced text understanding capabilities.

The Lumina-T2X primarily supports flow matching but also supports denoising probabilistic models (DDPM), as most algorithms are designed to be compatible with DDPM. Furthermore, it supports SD-1.5 and SDXL VAE. The latent space of SD-1.5 VAE can simultaneously encode image and video features, whereas SDXL offers superior visual quality but does not support video generation. Other configurations, such as RoPE, image augmentation policy, and generation, are fixed to be 1D, resize-then-crop, and parallel generation, respectively. The next version of Lumina-T2X will further explore these factors in depth.

## E    LUMINA-T2X SYSTEM

In this section, we introduce the family of Lumina-T2X, including Lumina-T2I, Lumina-T2V, Lumina-T2MV, and Lumina-T2Speech. For each modality, Lumina-T2X is independently trained with diverse configurations optimized for varying scenarios, such as different text encoders, VAE latent spaces, and parameter sizes. The detailed configurations are provided in Appendix D. Lumina-T2I is the key component of our Lumina-T2X system, where we utilize the T2I task as a testbed for validating the effectiveness of each component discussed in Section 3.2.  Notably, our most advanced Lumina-T2I model with a 5B Flag-DiT, 7B LLaMA text encoder, and SDXL latent space demonstrates superior visual quality and accurate text-to-image alignment. Then, we can extend the explored architecture, hyper-parameters, and other training details to videos, multi-views, and speech generation. Since videos and multi-views of 3D objects usually contain up to 1 million tokens, Lumina-T2V and Lumina-T2MV adopt a 2B Flag-DiT, CLIP-L/G text encoder, and SD-1.5 latent space. Although this configuration slightly reduces visual quality, it provides an effective balance for processing long sequences and a joint latent space for images and videos. Motivated by previous approaches (Ho et al., 2022a; Chen et al., 2023b), Lumina-T2I, Lumina-T2V, and Lumina-T2MV employ a multi-stage training approach, starting from low-resolution, short-duration data while ending with high-resolution, long-duration data. Such a progressive training strategy significantly improves the convergence speed of Lumina-T2X. For Lumina-T2Speech, since the feature space of the spectrogram shows a completely different distribution than images, we directly tokenize the spectrogram without using a VAE encoder and train a randomly initialized Flag-DiT conditioned on a phoneme encoder for T2Speech generation.

## F    ADVANCED APPLICATIONS OF LUMINA-T2X

During inference, Lumina-T2X supports various advanced applications including resolution extrapolation, style-consistent generation, compositional generation, and high-resolution editing. All of these applications can be achieved in a training-free manner integrated into a unified framework as illustrated in Figure 5. In this section, we provide implementing details of each application.

### F.1    UNLEASHING THE FULL POTENTIAL OF LUMINA-T2X WITH RESOLUTION EXTRAPOLATION

**Direct Resolution Extrapolation**    The simplest way to achieve resolution extrapolation is by increasing the sequence length and repositioning the `[nextline]` token. This allows Lumina-T2X to infer at higher resolutions than those used during training. Ideally, this should work well – because RoPE encodes relative positions rather than absolute positions, and its characteristic of long-term decay (Su et al., 2024) can mitigate the negative effects of unseen context lengths.

However, in practice, we find that the effects of direct resolution extrapolation are very limited, and the model quickly collapses after a certain degree of extrapolation. This echoes the findings on LLMs with RoPE — the long-range decay of RoPE is insufficient to suppress the anomalies brought about by unseen context lengths (Chen et al., 2023d). Although Position Interpolation (PI) is proposed in Chen et al. (2023d) to improve context length generalizability, fine-tuning is still necessary.

**NTK-Aware Scaled RoPE**  Using the transformer architecture and 1D RoPE (Su et al., 2024), Lumina-T2X can seamlessly integrate the context window extension methods designed for LLMs to achieve inference-time extrapolation.

RoPE encodes position information with a frequency matrix $\Theta = \text{Diag}(\theta_0, \cdots, \theta_d, \cdots, \theta_{|D|/2-1})$ with $\theta_d = b^{-2d/|D|}$, where $b$ is the rotary base. Following NTK-aware scaled RoPE (loc, 2024), when performing resolution extrapolation, we scale the rotary base $b$ such that the lowest frequency term is equivalent to PI, allowing a gradual transition from position extrapolation of high-frequency terms to position interpolation of low-frequency ones, achieving tuning-free generalization from the training context length $L$ to the testing context length $L'$. For any scale factor $s = L'/L$ ($L' > L$), the scaled base can be expressed as $b' = b \cdot s^{\frac{|D|}{|D|-2}}$. Such a simple operation allows Lumina-T2X to extrapolate to $\sim 3\times$ context length (1.8K images).

**Time Shifting**  We look into the discretization of time schedule to solve the Flow ODE during sampling, which is of vital importance in controlling the denoising rate. A common approach is to use Euler's method with a constant step size. However, similar to the observation in diffusion schedules (Teng et al., 2023; Hoogeboom et al., 2023; Hwang et al., 2024), we found that the high-resolution images are less corrupted and retain the global structure for a wider range of time under the linear interpolation schedule in flow matching.

This observation motivates us to adjust the time discretization schedule for high-resolution image generation to match the corresponding schedule of origin resolution. More specifically, the low-resolution image at time $t$ is defined as $x_t^{\text{low}} = tx^{\text{low}} + (1-t)\epsilon$, while the high-resolution image is $x_t^{\text{high}} = tx^{\text{high}} + (1-t)\epsilon$. To compare their noise strength at the same resolution, we downsample $x^{\text{high}}$ $m$ times with average pooling to match the lower resolution. The downsampled image is $x_t^{\text{high}} = tx^{\text{low}} + \frac{(1-t)}{m}\epsilon$, with the variance of Gaussian noise reduced to $1/m$ using average pooling due to the central limit theorem. The signal-to-noise ratio (SNR) become $m^2$ times larger, since

$$\text{SNR}^{\text{high}} = \frac{m^2 t^2}{(1-t)^2} = m^2 \text{SNR}^{\text{low}}. \tag{10}$$

Therefore, we can match their SNR by shifting the timestep of the high-resolution image, following

$$\frac{m^2 t_{\text{high}}^2}{(1-t_{\text{high}})^2} = \frac{t_{\text{low}}^2}{(1-t_{\text{low}})^2}, \tag{11}$$

and we can write the exact shifted timestep by simplifying the above equation

$$t_{\text{high}} = \frac{t_{\text{low}}}{m - mt_{\text{low}} + t_{\text{low}}}. \tag{12}$$

This coincides with the Time Shifting schedule in (Esser et al., 2024) and other counterparts in diffusion literature (Hoogeboom et al., 2023; Hwang et al., 2024). However, in practice, we find that setting $m$ to a larger value than the resolution scaling constant can further boost the quality of generated images. We visualize generated images using different shifting values under different resolutions in Figure 9 and adopt $m = 6.0$ in all the experiments.

**Proportional Attention**  During resolution-extrapolation, the sequence length is significantly greater than that during training. With longer input sequences, the attention module tends to aggregate information across a wider range of context tokens. This gap between training and inference leads the model to generate high-resolution images containing repeated, incomplete, and disordered patterns. To make up for this, we can scale each term in the attention softmax by a constant $c$, named proportional attention. This operation restricts the model to concentrate on fewer context tokens, which is similar to the training resolution. To determine the best value of $c$, we adopt the setting

in (Jin et al., 2024) where they start from the entropy perspective and find that the attention entropy also varies in proportion to resolutions. They set this hyper-parameter as $c = \sqrt{\log_{L_{\text{train}}} L_{\text{infer}}}$ to mitigate entropy fluctuation, where $L_{\text{train}}$ and $L_{\text{infer}}$ are the numbers of tokens during training and inference, respectively. The final formulation of our proportional attention is:

$$A = \text{softmax}(\frac{QK^T}{\sqrt{d}} \sqrt{\log_{L_{\text{train}}} L_{\text{infer}}}). \tag{13}$$

**Relationship with Other Resolution Extension Methods** Due to the enormous computational cost of high-resolution models and the scarcity of high-resolution image data, directly training high-resolution generative models is costly. Therefore, high-resolution fine-tuning/adaptation (Zheng et al., 2024a; Cheng et al., 2024; Guo et al., 2024; Chen et al., 2024b) and tuning-free high-resolution generation (He et al., 2023; Du et al., 2024; Haji-Ali et al., 2023; Hwang et al., 2024; Huang et al., 2024) are the mainstream choices today. Among the tuning-free approaches, DemoFusion (Du et al., 2024), ElasticDiffusion (Haji-Ali et al., 2023), and Upsample Guidance (Hwang et al., 2024) operate in a model-agnostic manner, while ScaleCrafter (He et al., 2023) and FouriScale (Huang et al., 2024) apply the dilated convolution mechanism specifically tailored to the CNN-based diffusion models. In this paper, we explore the tuning-free resolution extrapolation potential of Lumina-T2X from the perspectives of Flow-based Diffusion Transformers with RoPE, an area not extensively studied within the field. Different from previous approaches, which either require computationally demanding fine-tuning with expensive high-resolution images or complex architecture/pipeline modifications over pre-trained models, Lumina-T2X can generate high-resolution images simply by repositioning the `[nextline]` tokens to the specific slot.

## F.2 STYLE-CONSISTENT GENERATION

The transformer-based diffusion model architecture makes Lumina-T2I naturally suitable for self-attention manipulation applications like style-consistent generation. A representative approach is shared attention (Hertz et al., 2023), which enables generating style-aligned batches without specific tuning of the model. Specifically, it uses the first image in a batch as the anchor/reference image, allowing the queries from other images in the batch to access the keys and values of the first image during the self-attention operation. This kind of information leakage effectively promotes a consistent style across the images in a batch. Typically, this can be achieved by concatenating the keys and values of the first image with those of other images before self-attention. However, in diffusion transformers, it is important to note that keys from two images contain duplicated positional embeddings, which can disrupt the model's awareness of spatial structures. Therefore, we need to ensure that key/value sharing occurs before RoPE, which can be regarded as appending a reference image sequence to the target image sequence.

## F.3 COMPOSITIONAL GENERATION

Compositional, or multi-concepts text-to-image generation (Jiménez, 2023; Bar-Tal et al., 2023; Yang et al., 2024), which requires the model to generate multiple subjects at different regions of a single image, is seamlessly supported by our transformer-based framework. Users can define $N$ different prompts and $N$ bounding boxes as masks for corresponding prompts. Our key insight is to restrict the cross-attention operation of each prompt within the corresponding region during sampling. More specifically, at each timestep, we crop the noisy data $x_t$ using each mask and reshape the resulting sub-regions into a sub-region batch $\{x_t^1, x_t^2, \dots, x_t^N\}$, corresponding to the prompt batch $\{y^1, y^2, \dots, y^N\}$. Then, we compute cross-attention using this sub-region batch and prompt batch and manipulate the output back to the complete data sample. We only apply this operation to cross-attention layers to ensure the text information is injected into different regions while keeping the self-attention layers unchanged to ensure the final image is coherent and harmonic. We additionally set the global text condition as the embedding of the complete prompt, i.e., concatenation of all prompts, to enhance global coherence.

## F.4 HIGH-RESOLUTION EDITING

Beyond high-resolution generation, our Lumina-T2I can also perform image editing (Hertz et al., 2022; Brooks et al., 2023), especially for high-resolution images. Considering the distinct features of

Table 4: We compare the training setups of Lumina-T2I with PixArt-$\alpha$. Lumina-T2I is trained purely on 14 million filtered high-quality (HQ) image-text pairs, whereas PixArt-$\alpha$ benefits from an additional 11 million high-quality natural image-text pairs. Remarkably, despite having 8.3 times more parameters, Lumina-T2I only incurs 35% of the computational costs compared to PixArt-$\alpha$-0.6B.

| PixArt-$\alpha$-0.6B with T5-3B | | | | | Lumina-T2I-5B with LLaMA-7B | | | | |
|---|---|---|---|---|---|---|---|---|---|
| Res. | #images | Batch Size | Learning Rate | GPU days (A100) | Res. | #images | Batch Size | Learning Rate | GPU days (A100) |
| 256 | 1M ImageNet | 1024 | $2\times10^{-5}$ | 44 | - | - | - | - | - |
| 256 | 10M SAM | 11392 | $2\times10^{-5}$ | 336 | - | - | - | - | - |
| 256 | 14M HQ | 11392 | $2\times10^{-5}$ | 208 | 256 | 14M HQ | 512 | $1\times10^{-4}$ | 96 |
| 512 | 14M HQ | 2560 | $2\times10^{-5}$ | 160 | 512 | 14M HQ | 256 | $1\times10^{-4}$ | 96 |
| 1024 | 14M HQ | 384 | $2\times10^{-5}$ | 80 | 1024 | 14M HQ | 128 | $1\times10^{-4}$ | 96 |

different editing types, we first classify image editing into two major categories, namely style editing and subject editing. For style editing, we aim to change or enhance the overall visual style, such as color, environment, and texture, without modifying the main object of the image, while subject editing aims to modify the content of the main object, such as addition, replacement, and removal, without affecting the overall visual style. Then, we leverage a simple yet effective method to achieve this image editing within the Lumina-T2I framework. Specifically, given an input image, we first encode it into latent space using the VAE encoder and interpolate the image latent with noise to get the intermediate noisy latent at time $\lambda$. Then, we can solve the Flow ODE from $\lambda$ to $1.0$ with desired prompts for editing as text conditions. Due to the powerful generation capability of our model, it can faithfully perform the ideal editing while preserving the original details in high resolution. However, in style editing, we find that the mean and variance are highly correlated with image styles. Therefore, the above method still suffers from style leakage since the interpolated noisy data still retains the style of the original image in its mean and variance. To eliminate the influence of the original image styles, we perform channel-wise normalization on input images, transforming them to zero mean and unit variance.

## G  ADDITIONAL EXPERIMENTAL RESULTS

### G.1  RESULTS OF TEXT-TO-IMAGE GENERATION

Figure 7 and Figure 8 showcase more text-to-image generation samples of our Lumina-T2I. Besides, we provide the comparison of training setups between Lumina-T2I and PixArt-$\alpha$ in Table 4.

### G.2  INFLUENCE OF TIME SHIFTING

As mentioned before, time shifting is critical to generate images with higher resolution than training. We explore the impact of different values of the shifting factor $m$. As depicted in Figure 9, it is surprising that a larger value of $m$ significantly improves the overall visual quality for nearly all resolutions, ranging from 256 to 2048. When scaling this shifting factor from 1.0 to 10.0, the main subject in the image becomes closer and brighter, exhibiting fewer artifacts. We speculate this is because a larger $m$ indicates spending more steps at the early stage of sampling, which is important for the diffusion model to compose the global structure layout. In contrast, we can skip some steps at the end of sampling since the model is performing an easier task similar to pure denoising.

**Style-Consistent Generation**   Batch generation of style-consistent content holds immense value for practical application scenarios (Hertz et al., 2023; Tewel et al., 2024). Here, we demonstrate that through simple key/value information leakage, Lumina-T2I can generate impressive style-aligned batches. As shown in Figure 10, leveraging a naive attention-sharing operation, we can observe strong consistency within the generated batches. Thanks to the full-attention model architecture, we can obtain results comparable to those in (Hertz et al., 2023) without using any tricks such as Adaptive Instance Normalization (AdaIN) (Huang & Belongie, 2017). Furthermore, we believe that, as previous arts (Hertz et al., 2023; Tewel et al., 2024) illustrate, through appropriate inversion

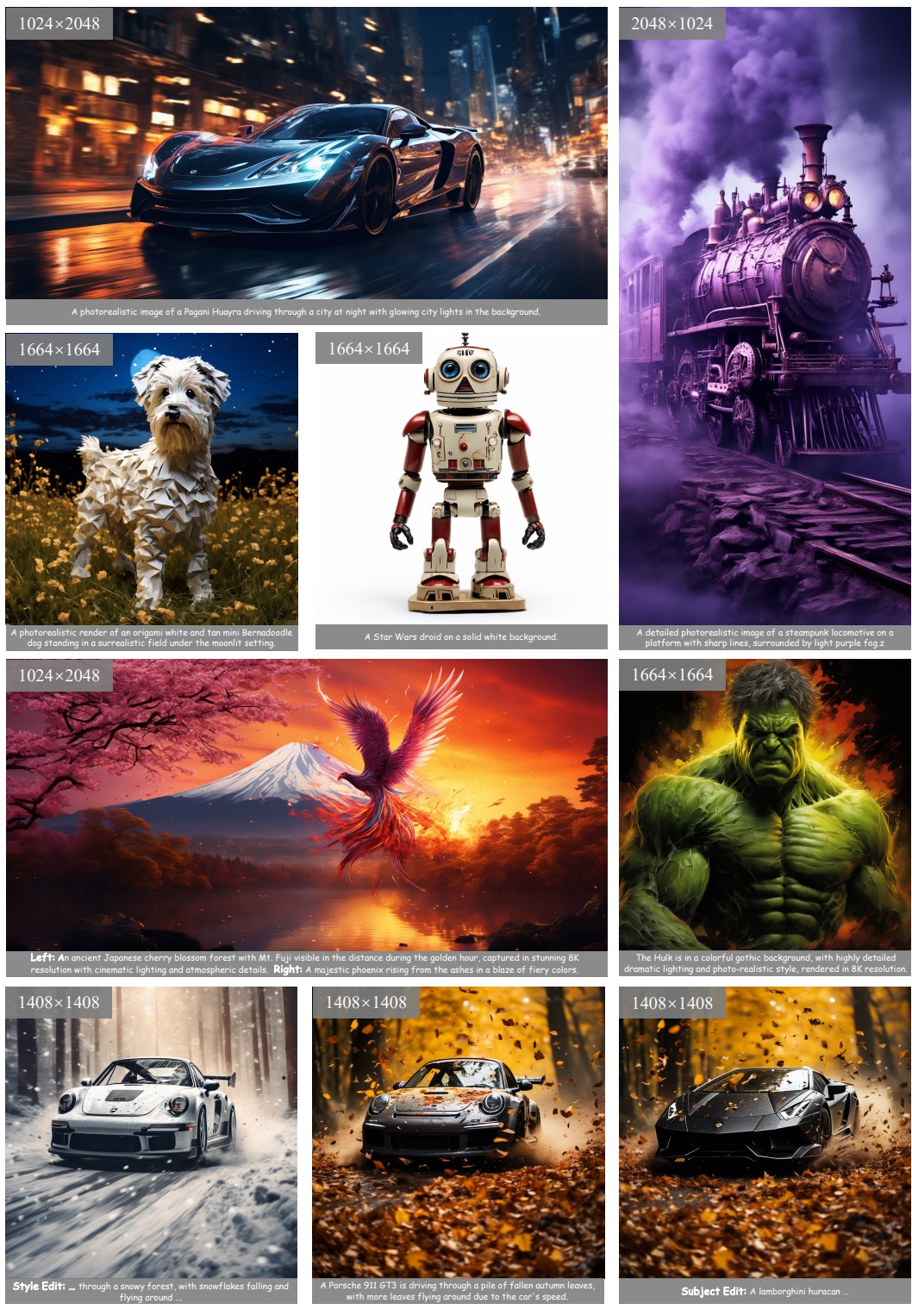

Figure 7: Lumina-T2I is capable of generating higher-resolution images than its training resolution (1024 × 1024), producing photorealistic images at arbitrary resolutions and aspect ratios. Additionally, it can compose images based on multiple captions (third row), perform seamless high-resolution editing to image styles or subjects (last row), and support a diverse range of topics and styles for image generation.

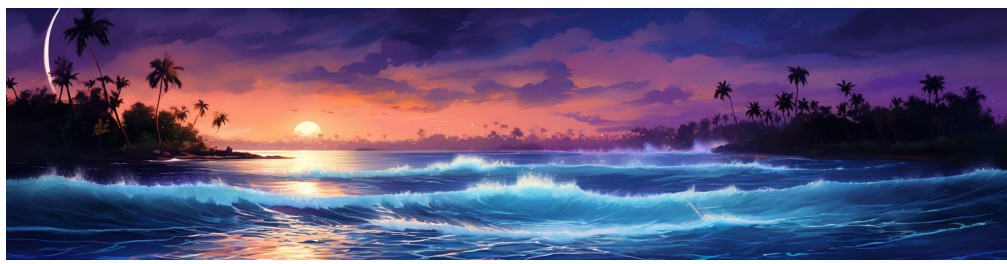

A serene twilight beach scene with silhouetted palm trees and bioluminescent waves, digital oil painting.

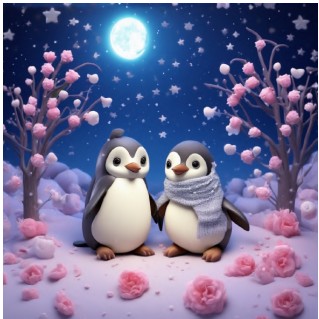
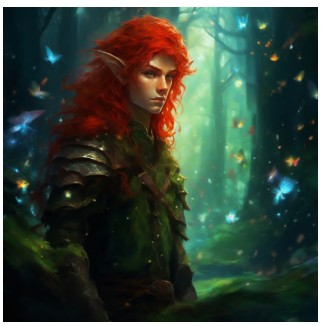
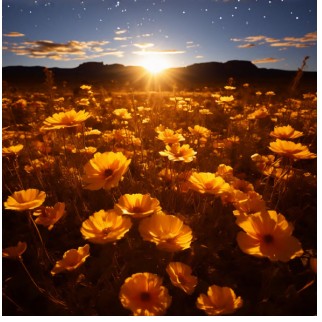

Two cute penguins in a romantic Valentine's yarn setting under the moonlight with pastel colors.

A red-haired male elf hunter with a shy expression is standing in a mystical forest, surrounded by fairy tale-like elements and vibrant spectral colors.

A photograph showcases the beauty of desert flowers and mirrors illuminated by the soft morning light. The image, extremely photorealistic and meticulously detailed, depicts a lonely desert atmosphere with stars shining overhead.

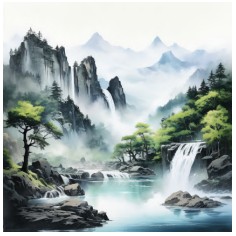
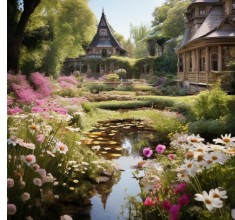
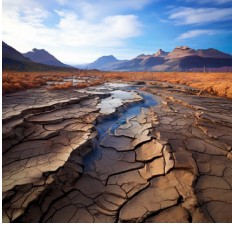
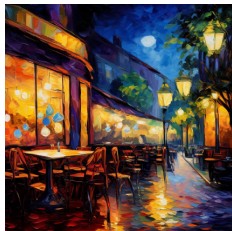

A serene mountain landscape in the style of a Chinese ink painting, with a waterfall cascading down into a crystal-clear lake surrounded by ancient pines.

A beautiful Victorian-era botanical garden featuring a charming pond and lovely daisies.

A realistic landscape shot of the Northern Lights dancing over a snowy mountain range in Iceland, with long exposure to capture the motion and vibrant colors.

An impressionist painting of a bustling café terrace at night, with vivid colors and lively brush strokes.

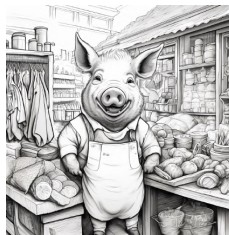
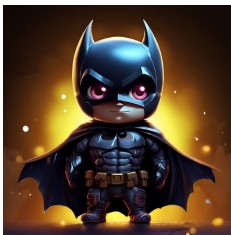
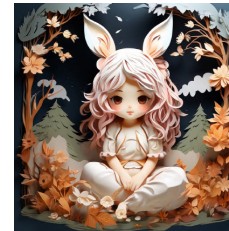
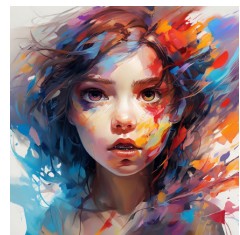

Detailed pen and ink drawing of a happy pig butcher selling meat in its shop.

Batman, cute modern Disney style, ultra-detailed, gorgeous, trending on dribble

A detailed paper cut craft and illustration of a cute anime bunny girl sitting in the woods.

A young girl's face disintegrates while beautiful colors fill her features, depicted in fluid dynamic brushwork with colorful dream-like illustrations.

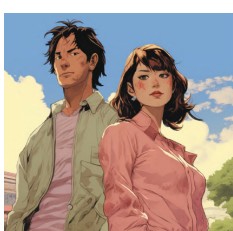
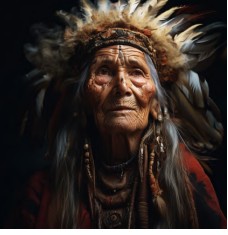
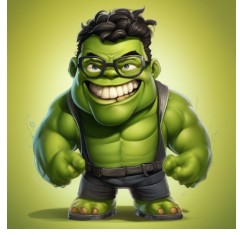
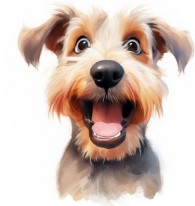

An 80s anime still illustrated, featuring a man and a woman in a city park, wearing retro clothing with muted pastel colors.

An old shaman woman adorned with feathers and leather, portrayed in a photorealistic illustration with soft lighting and sharp focus.

An anthropomorphic Hulk, wearing glasses and smiling, is depicted in a cute and funny character design

a watercolor portrait of a Terrier dog, smiling and making a cute facial expression while looking at the camera, in Pixar style.

Figure 8: Lumina-T2I is capable of generating images with arbitrary aspect ratios, delivering superior visual quality and fidelity while adhering closely to given text instructions.

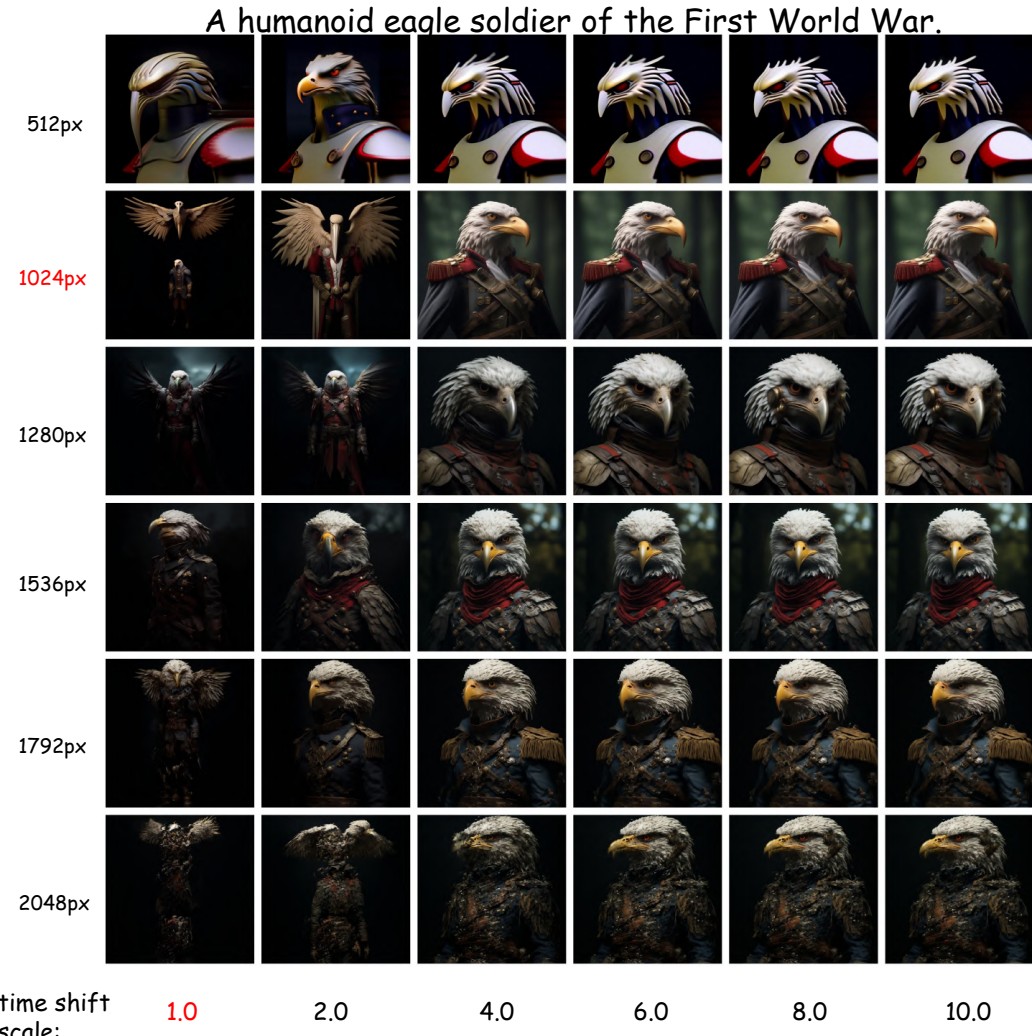

Figure 9: Qualitative effects of time shifting on various resolutions. A larger Time Shifting scale effectively improves the visual quality of generated images.

techniques, we can achieve style/concept personalization at zero cost, which is a promising direction for future exploration.

**Compositional Generation**    As illustrated in Figure 11, we present demos of compositional generation (Yang et al., 2024; Bar-Tal et al., 2023) using the method described in Appendix F.3. We can define an arbitrary number of prompts and assign each prompt an arbitrary region. Lumina-T2I successfully generates high-quality images in various resolutions that align with complex input prompts while retaining overall visual coherence. This demonstrates that the design choice of our Lumina-T2I offers a flexible and effective method that excels in generating complex high-resolution multi-concept images.

**High-Resolution Editing**    Following the methods outlined in Appendix F.4, we perform style and subject editing on high-resolution images (Hertz et al., 2022; Brooks et al., 2023; Kawar et al., 2023; Sheynin et al., 2023). As depicted in Figure 12, Lumina-T2I can seamlessly modify global styles or add subjects without the need for additional training. Furthermore, we analyze various factors such as starting time and latent feature normalization in image editing, as shown in Figure 13. By varying the starting time from 0 to 1, we find that a starting time near 0 leads to complete spatial misalignment,

A winter forest…     A sleigh…     A snowman…     A snowy cabin…

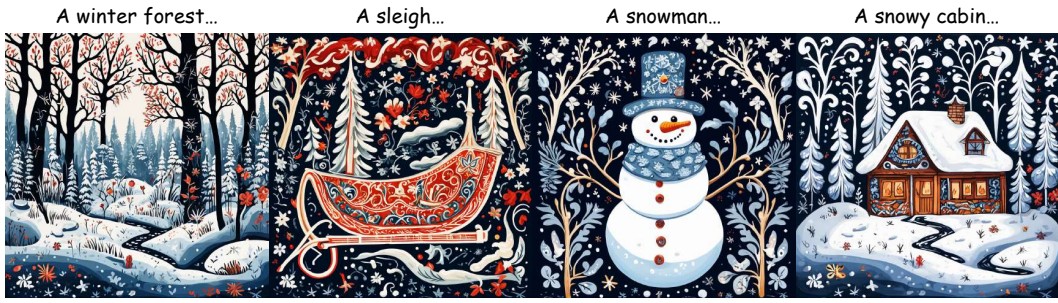

…in Scandinavian folk art style.

Candles and roses…     A bottle…     A pizza…     A chef…

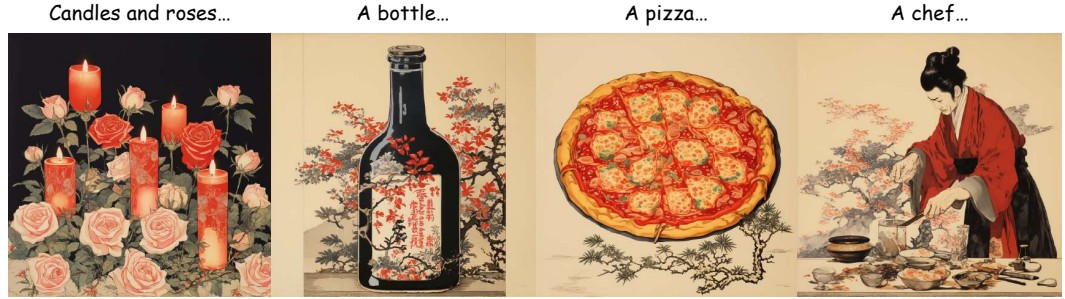

…in Japanese ukiyo-e style.

Toy train…     Toy airplane…     Toy bicycle…     Toy car…

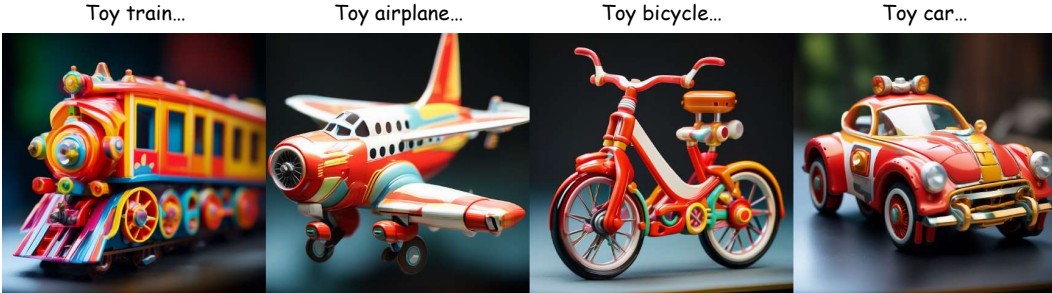

…colorful, macro photo.

Figure 10: Style-consistent image generation samples produced by Lumina-T2I. Given a shared style description, Lumina-T2I can generate a batch of images with diverse style-consistent contents.

while a starting time near 1 results in unchanged content. Setting the starting time to 0.2 provides a good balance between adhering to the editing instructions and preserving the structure of the original image. Compared with the generated image without normalization, it is clear that channel-wise normalization can effectively remove the original style of the input image while preserving its main content. By normalizing the latent features of the original image, our approach to image editing can better handle the editing instructions.

**Comparison with PixArt-$\alpha$** Compared to PixArt-$\alpha$ (Chen et al., 2023b), Lumina-T2I can generate images at resolutions ranging from $512^2$ pixels to $1792^2$ pixels. As demonstrated in Figure 14, PixArt-$\alpha$ struggles to produce high-quality images at both lower and higher resolutions than the size of images used during training. Lumina-T2I utilizes RoPE, the [nextline] token, as well as layer-wise relative position injection, enabling it to effectively handle a broader spectrum of resolutions. In contrast, PixArt-$\alpha$ relies on absolute position embedding and limits positional information to the initial layer, leading to a degradation in performance when generating images at out-of-distribution scales.

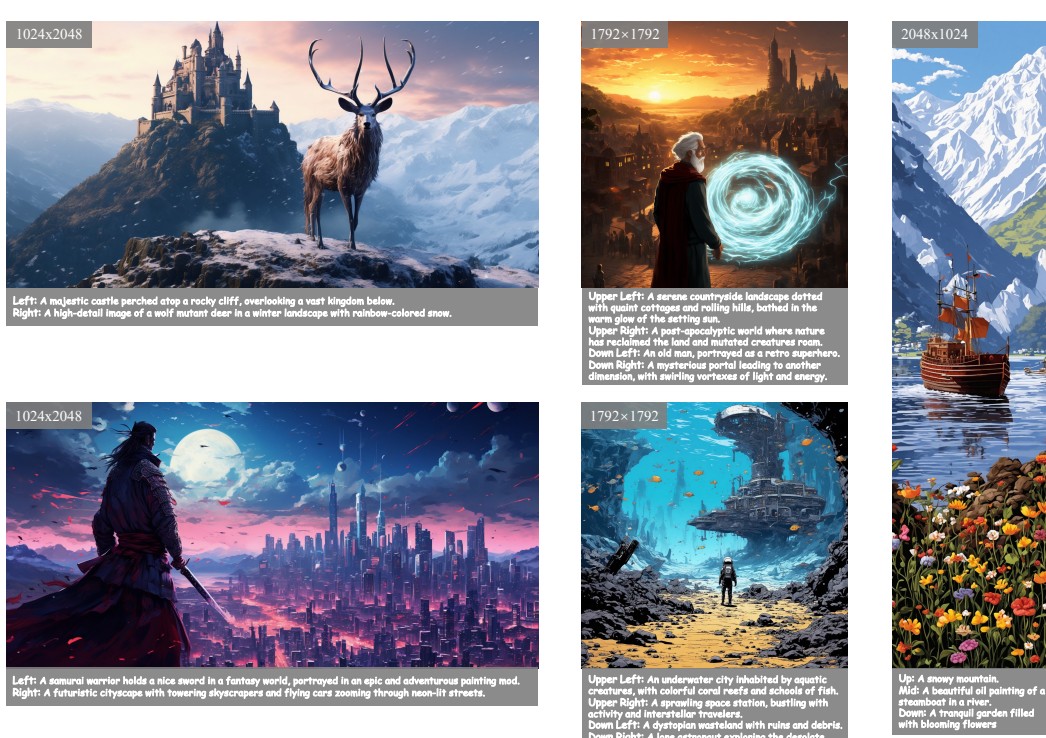

Figure 11: Compositional generation samples of Lumina-T2I. Our Lumina-T2I framework can generate high-quality images with intricate compositions based on a combination of prompts and

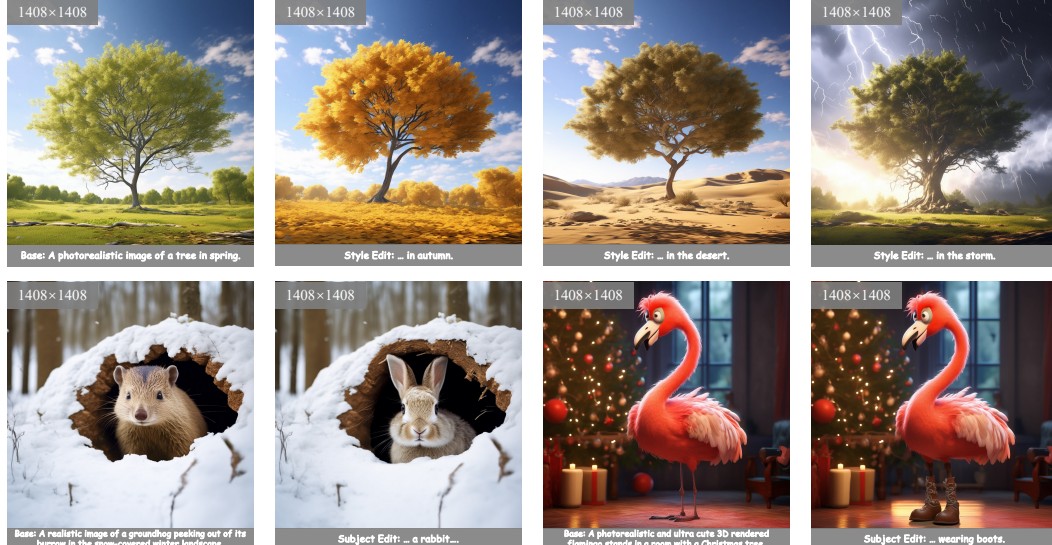

Figure 12: Demonstrations of style editing and subject editing over high-resolution images in a training-free manner.

Apart from resolution extrapolation, Lumina-T2I also adopts a simplified training pipeline, as shown in Table 4. Ablation studies conducted on ImageNet indicate that training with natural image domains such as ImageNet results in higher training losses in subsequent stages. This suggests that synthetic images from JourneyDB and natural images collected online (*e.g.*, *LAION* (Schuhmann et al., 2021; 2022), *COYO* (Byeon et al., 2022), *SAM* (Kirillov et al., 2023), and *ImageNet* (Deng et al., 2009)) belong to distinct distributions. Motivated by this observation, Lumina-T2I trains directly on

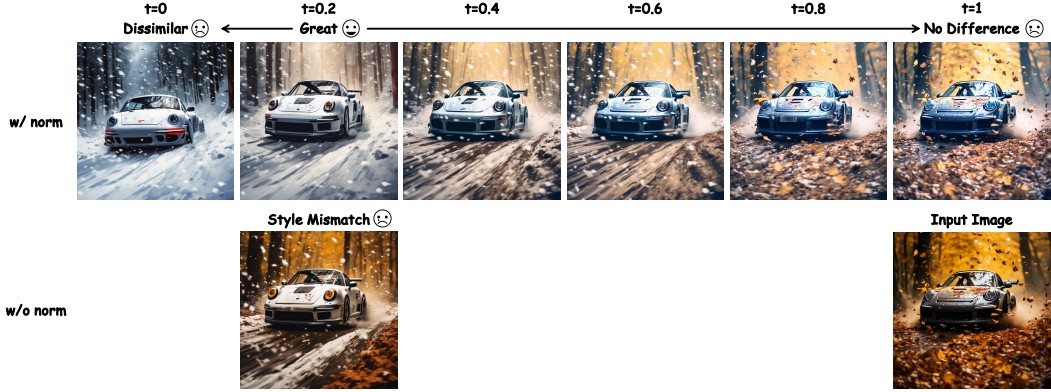

Figure 13: Qualitative effects of the starting time and latent feature normalization in style editing. A starting time near 0.2 yields a good balance between preserving the original content and incorporating the desired target style, while removing normalization greatly hinders the model's ability to effectively transform image styles.

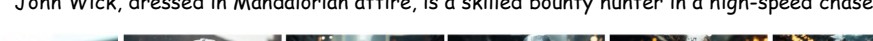

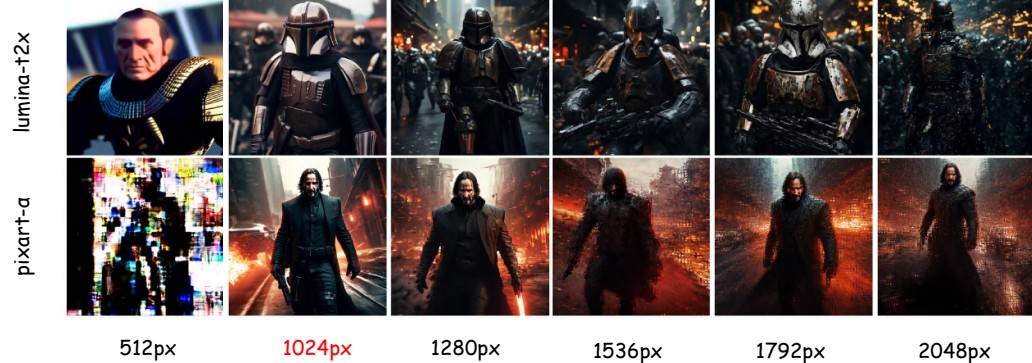

Figure 14: Qualitative comparison between Lumina-T2I and PixArt-$\alpha$ in generating images at multiple resolutions. The samples from Lumina-T2I demonstrate better alignment with the given text and superior visual quality across all resolutions compared to those from PixArt-$\alpha$.

high-resolution synthetic domains to reduce computational costs and avoid suboptimal initialization. Additionally, inspired by the fast convergence of the FID score observed when training on ImageNet, Lumina-T2I adopts a 5 billion Flag-DiT, which has 8.3 times more parameters than PixArt-$\alpha$, yet incurs only 35% training costs (288 A100 GPU days compared to 828 A100 GPU days).

**Analysis of Gate Distribution in Zero-Initialized Attention**   Cross-attention (Tang et al., 2022; Blattmann et al., 2023a) is the de-facto standard for text conditioning. Unlike previous methods, Lumina-T2I employs zero-initialized attention mechanism (Gao et al., 2023; Zhang et al., 2023b), which incorporates a zero-initialized gating mechanism to adaptively control the influence of text-conditioning across various heads and layers. Surprisingly, we observe that zero-initialized attention can induce extremely high levels of sparsity in text conditioning. As shown in Figure 15(a), we visualize the gating values across heads and layers, revealing that most gating values are close to zero, with only a small fraction exhibiting significant importance. Interestingly, the most crucial text-conditioning heads are predominantly found in the middle layers, suggesting that these layers play a key role in text conditioning. To consolidate this observation, we truncated gates below a certain threshold and found that 80% of the gates can be deactivated without affecting the quality of image generation, as demonstrated in Figure 15(b). This observation suggests the possibility of truncating most cross-attention operations during sampling, which can greatly reduce inference time.

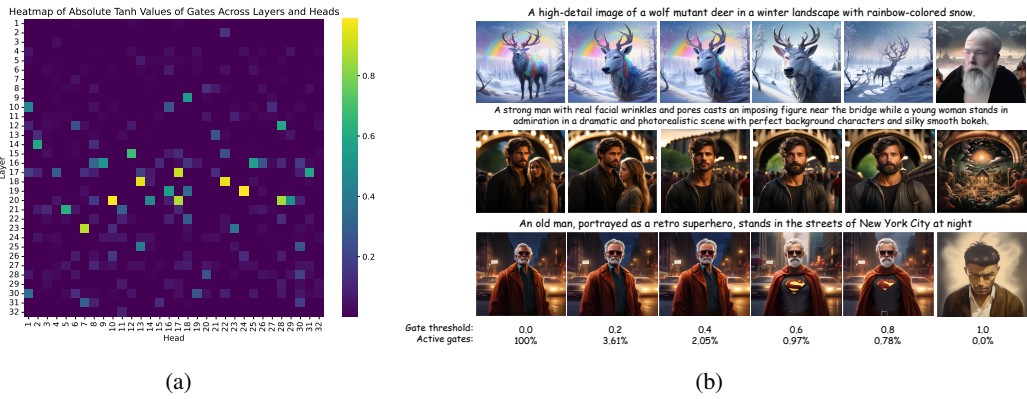

Figure 15: Gated cross-attention in Lumina-T2I. (a) Absolute tanh values of all gates across all layers and heads. (b) Qualitative results of generated images under different gate thresholds.

# H APPLYING LUMINA-T2X TO MORE MODALITIES

## H.1 RESULTS OF LUMINA-T2V

**Basic Setups** Lumina-T2V shares the same architecture with Lumina-T2I except for the introduction of a `[nextframe]` token, which provides explicit information about temporal duration. By default, Lumina-T2V uses CLIP-L/G (Radford et al., 2021) as the text encoder and employs a Flag-DiT with 2 billion parameter as the diffusion backbone. Departing from previous approaches (Guo et al., 2023; Wang et al., 2023c; Jiang et al., 2023; Chen et al., 2023a; 2024a; Blattmann et al., 2023c; Zhou et al., 2023; Ho et al., 2022a; Blattmann et al., 2023b; Hu et al., 2023; Chen et al., 2023c; Wang et al., 2023b; Zhang et al., 2023c; Xing et al., 2023; Gupta et al., 2023; Wu et al., 2023) that rely on T2I checkpoints for T2V initialization and adopt decoupled spatial-temporal attention, Lumina-T2V takes a different route by initializing the Flag-DiT weights randomly and leveraging a full-attention mechanism that allows for interaction among all spatial-temporal tokens. Although this choice significantly slows down the training and overall inference speed, we believe that such an approach holds greater potential, particularly when ample computational resources are available.

Lumina-T2V is independently trained on a subset of the Panda-70M dataset (Chen et al., 2024c) and the collected Pexel dataset, comprising of 15 million and 40,000 videos, respectively. Similar to Lumina-T2I, Lumina-T2V employs a multi-stage training strategy that starts with shorter, low-resolution videos and subsequently advances to longer, higher-resolution videos. Specifically, in the initial stage, Lumina-T2V is trained on videos of a fixed size – such as 512 pixels in both height and width, and 32 frames in length for the Pexel dataset, which collectively comprise approximately 32,000 tokens. During the second stage, it learns to handle videos of varying resolutions and durations, while imposing a limit of 128,000 tokens to maintain computational feasibility.

**Observations of Lumina-T2V** We observe that Lumina-T2V with large batch size can converge, while a small batch size struggles to converge. As shown in Figure 16(a), increasing the batch size from 32 to 1024 leads to loss convergence. On the other hand, similar to the observation in ImageNet experiments, increasing model parameters leads to faster convergence in video generation. As shown in Figure 16(b), as the parameter size increases from 600M to 5B, we consistently observe lower loss for the same number of training iterations.

**Samples of Video Generation** As shown in Figure 17, the first stage of Lumina-T2V is able to generate short videos with scene dynamics such as scene transitions, although the generated videos are limited in terms of resolution and duration, with a maximum of 32K total tokens. After the second stage training on longer-duration and higher-resolution videos, Lumina-T2V can generate long videos with up to 128K tokens in various resolutions and durations. The generated videos, as illustrated in Figure 18, exhibit temporal consistency and richer scene dynamics, indicating a promising scaling trend when using more computational resources and data.

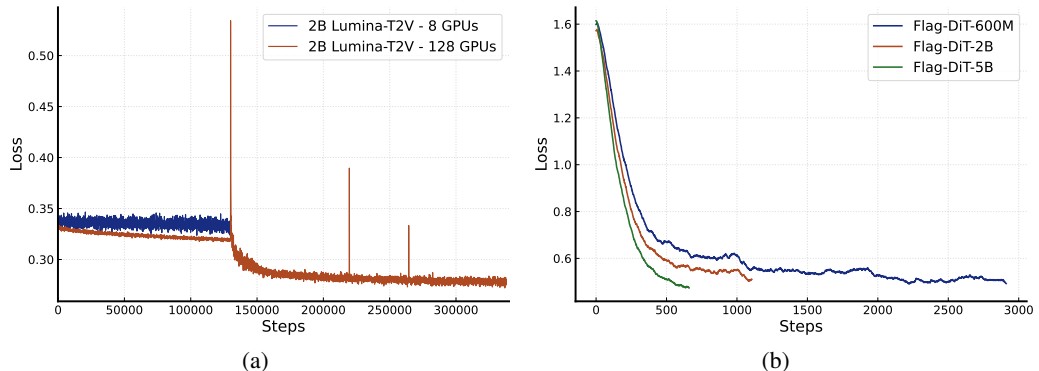

Figure 16: Training loss curve comparison between (a) 2B Flag-DiT trained on 8 GPUs and 128 GPUs, (b) different sizes of Flag-DiTs.

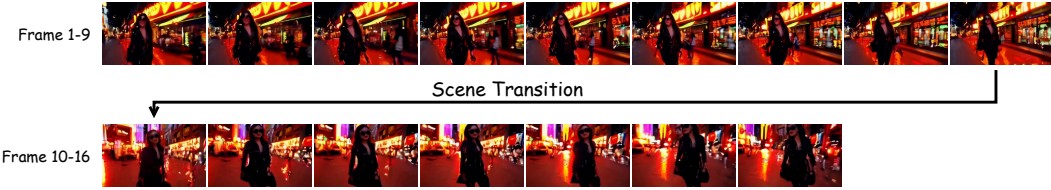

Figure 17: Short video generation samples of Lumina-T2V. Although the length and resolution of the generated videos are limited, these samples exhibit scene transition, indicating a promising way for long video generation.

## H.2 RESULTS OF LUMINA-T2MV

**Basic Setups** The multi-view images of a 3D object can be regarded as a distinct type of video format, emphasizing changes in the camera's position and orientation relative to a static object. We utilize multi-view images rendered from the Objaverse (Deitke et al., 2023) dataset to train a 5B Flag-DiT model with CLIP-L/G as the text encoder.

**Dataset** We employ the LVIS subset of the Objaverse dataset, which includes approximately 40K 3D objects. For textual prompts, we use the precise descriptions generated by Cap3D (Luo et al., 2024). For each object, we render 12 views around the object against a white background. The elevation is set at 30°, and the azimuth is uniformly distributed from 0° to 360°. We render the images at resolutions of $256 \times 256$ and $512 \times 512$, respectively, to train the 5B Flag-DiT model from scratch with different resolutions. Following Zero123++ (Shi et al., 2023a), we put the 12 rendered images into a single large image in the form of a $3 \times 4$ grid. The images are placed in row-wise order, with four images per row, across three rows. We do not fix the starting azimuth of the first image, only ensure that the azimuth of subsequent images increases sequentially by 30°. For twelve $256 \times 256$ multi-view images, this operation will result in a $1024 \times 768$ image, and so on for $512 \times 512$ images that will result in a $2048 \times 1536$ image.

**Training** We adopt a two-stage training strategy, starting with training on the $1024 \times 768$ images which are composed of twelve $256 \times 256$ images, and then training on the $2048 \times 1536$ images. During training, we provide only the merged 12-view images and corresponding text descriptions, without any information about camera parameters. The training is conducted on 16 NVIDIA A100 GPUs, each with 80GB of memory. For the low-resolution stage, we trained the Lumina-T2MV model with a batch size of 64 for 100K iterations, while for the high-resolutio n stage, we trained the Lumina-T2MV model with a batch size of 16 for 180K iterations. Other configurations are kept the same as the Lumina-T2I model.

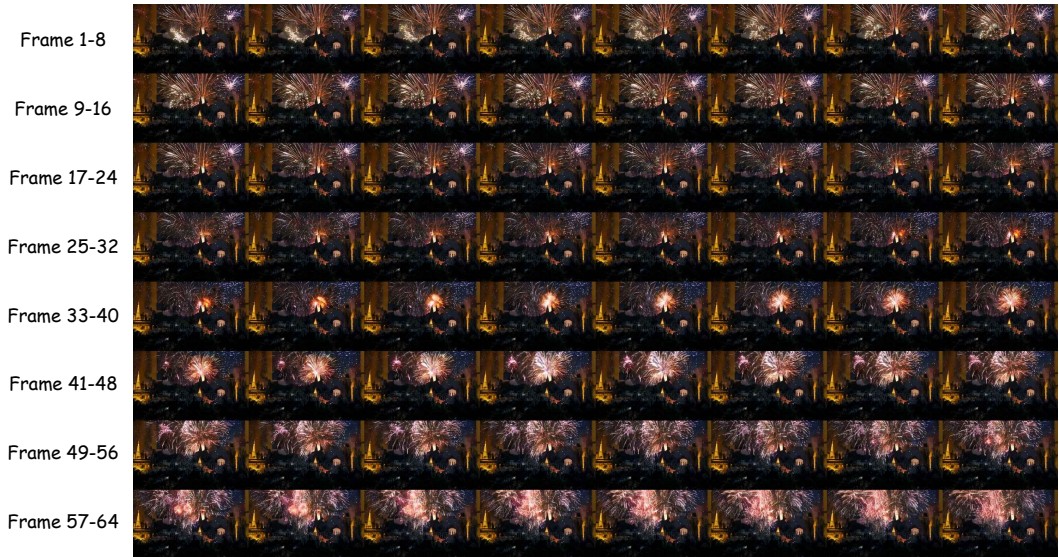

Fireworks over a Disney castle

Figure 18: Long video generation samples of Lumina-T2V. Lumina-T2V enables the generation of long videos with temporal consistency and rich scene dynamics.

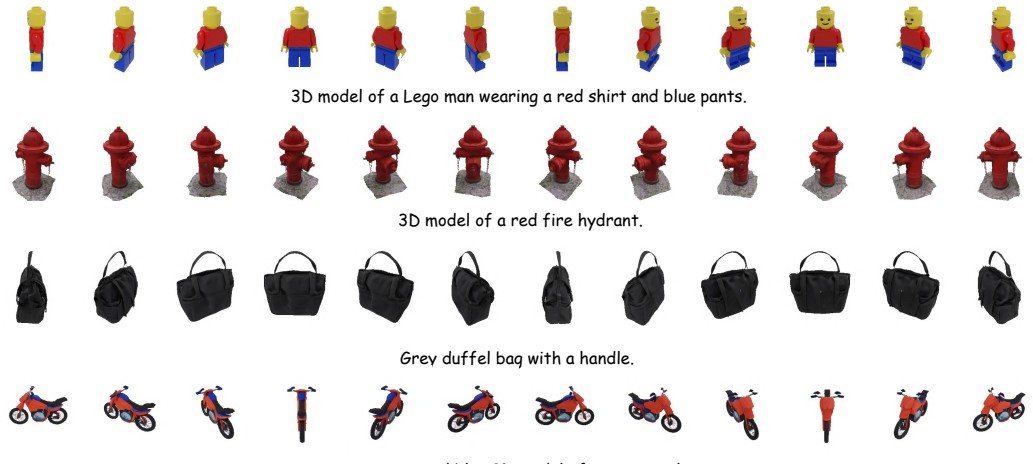

3D model of a Lego man wearing a red shirt and blue pants.

3D model of a red fire hydrant.

Grey duffel bag with a handle.

An orange and blue 3D model of a motorcycle.

Figure 19: Qualitative results of low-resolution multiview images generated by Lumina-T2MV

**Low-Resolution Multi-view Examples** The trained Flag-DiT model can generate twelve $256 \times 256$ images from different viewpoints based on the provided text prompt, demonstrating strong spatial consistency as shown in Figure 19. Although we did not provide the camera parameters, our model automatically understands the distribution of camera poses corresponding to different regions of the image and can generate reasonable multi-view images with viewpoint changes.

**High-Resolution Multi-view Examples** We observed that the model's capability to capture fine details of objects is limited by the $256 \times 256$ resolution of the first-stage training images. So we then use the $2048 \times 1536$ images for training, which are composed of twelve $512 \times 512$ images. Thanks to the powerful long-sequence modeling capability of our 5B Flag-DiT, the model maintains high performance as shown in Figure 20. Besides ensuring an accurate viewpoint of each generated multi-view image, we find a significant improvement in the quality of the generated details compared to the lower resolutions. We plan to scale up the training with more complex and denser camera

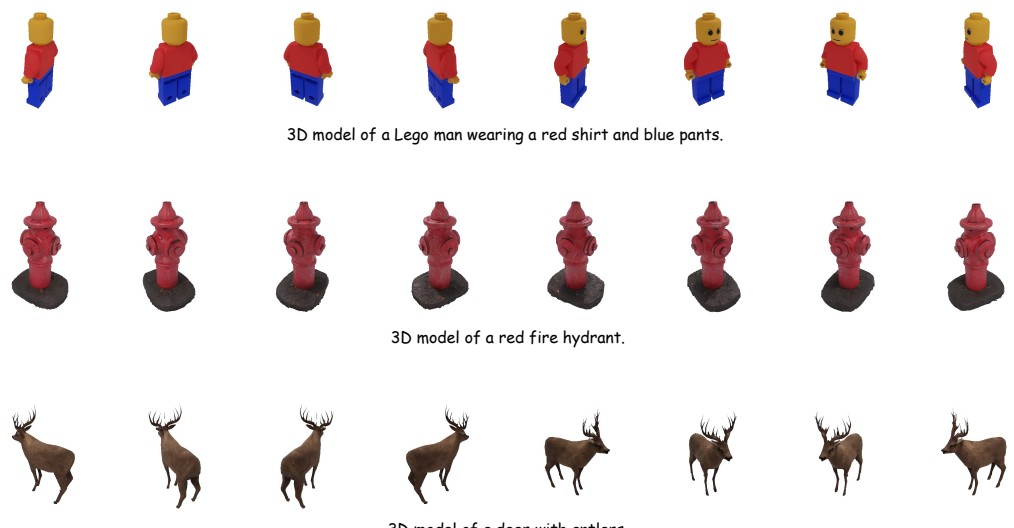

3D model of a Lego man wearing a red shirt and blue pants.

3D model of a red fire hydrant.

3D model of a deer with antlers.

Figure 20: Qualitative results of high-resolution multiview images generated by Lumina-T2MV

views, as well as higher image resolutions, to further explore the potential of our Lumina-T2MV model.

### H.3 RESULTS OF LUMINA-T2SPEECH

**Basic Setups** Lumina-T2Speech is also built on the Flag-DiT backbone consisting of a phoneme encoder and a pitch encoder. The size of the phoneme vocabulary is set as 73. In the pitch encoder, the size of the lookup table and encoded pitch embedding are set to 300 and 256, and the hidden channel is set to 256. We provide Lumina-T2Speech with different sizes of Flag-DiT following the configuration in the main text.

**Dataset** For a fair and reproducible comparison against other competing methods, we use the benchmark LJSpeech dataset (Ito, 2017). LJSpeech consists of 13,100 audio clips of 22050 Hz from a female speaker for about 24 hours in total. We convert the text sequence into the phoneme sequence with an open-source grapheme-to-phoneme conversion tool (Sun et al., 2019) [1]. Following the common practice (Chen et al., 2021; Min et al., 2021), we conduct preprocessing on the speech and text data: (1) extract the spectrogram with the FFT size of 1024, hop size of 256, and window size of 1024 samples; (2) convert it to a mel-spectrogram with 80 frequency bins; and (3) extract F0 (fundamental frequency) from the raw waveform using Parselmouth.

**Training** The Lumina-T2Speech has been trained for 200,000 steps using 1 NVIDIA 4090 GPU with a batch size of 64 sentences. The adam optimizer is used with $\beta_1 = 0.9, \beta_2 = 0.98, \epsilon = 10^{-9}$. We utilize HiFi-GAN (Kong et al., 2020) (V1) as the vocoder to synthesize waveform from the generated mel-spectrogram in all our experiments.

**Evaluation** We report word error rate (WER) to evaluate the intelligibility of speech by transcribing it using a whisper (Radford et al., 2023) ASR system following (Wang et al., 2023a). Style similarity (SIM) assesses the coherence of the generated speech in relation to the speaker's characteristics, and we employ the speaker verification model WavLM-TDNN (Chen et al., 2022) to evaluate the speaker similarity. We also conducted a crowd-sourced human evaluation via Amazon Mechanical Turk for Mean Opinion Score (MOS) test following (Protasio Ribeiro et al.), which is reported with 95% confidence intervals.

---

[1] https://github.com/Kyubyong/g2p

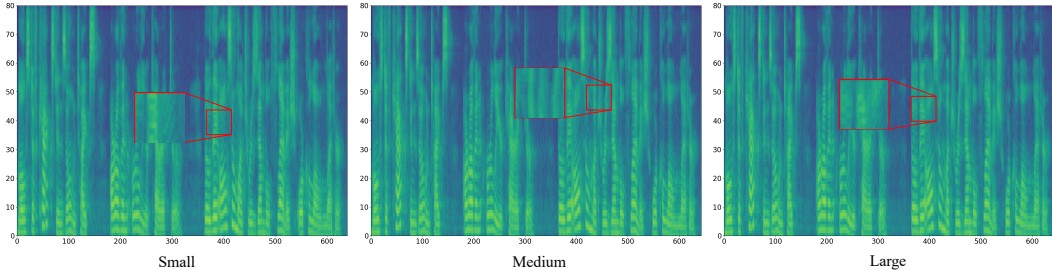

| Small | Medium | Large |

Figure 21: Visualizations of the reference and generated mel-spectrograms. The corresponding texts of generated speech samples is "*Most of Caxton's own types are of an earlier character, though they also much resemble Flemish or Cologne letter.*"

Table 5: Comparisons with existing text-to-speech methods and different configurations of Flag-DiT.

| Method | MOS | WER | SIM |
|---|---|---|---|
| GT | 4.34±0.07 | / | / |
| GT (voc.) | 4.18±0.05 | 5.3 | 99.2 |
| FastSpeech 2 (Ren et al., 2020) | 3.83±0.08 | / | / |
| DiffSpeech (Liu et al., 2022a) | 3.92±0.06 | / | / |
| WaveGrad (Chen et al., 2020b) | 4.00±0.00 | / | / |
| FastDiff 2 (Huang et al., 2023b) | 4.12±0.08 | / | / |
| Flag-DiT-S | 3.92±0.07 | 6.8 | 97.5 |
| Flag-DiT-B | 3.98±0.06 | 6.4 | 98.0 |
| Flag-DiT-L | 4.02±0.08 | 6.2 | 98.3 |
| Flag-DiT-XL | 4.01±0.07 | 6.3 | 98.4 |

**Results** We first evaluate the performance of FlagDiT with different configurations. The results have been shown in Table 5. Increasing the depth and number of layers in the transformer can significantly enhance the performance of the diffusion model, resulting in an improvement in both objective metrics and subjective metrics, which demonstrates that expanding the model size enables finer-grained room acoustic modeling.

For the intelligibility of the generated speech and style similarity, our Flag-DiT synthesizes accessible speech with good quality. For subjective evaluation, we compare our Flag-DiT with several text-to-speech approaches, including FastSpeech 2 (Ren et al., 2020), DiffSpeech (Liu et al., 2022a), WaveGrad (Chen et al., 2020b), and FastDiff 2 (Huang et al., 2023b). The superior results when compared to other diffusion-based approaches, such as DiffSpeech and WavGrad, indicate FlagDiT's potential in effectively modeling audio signals without any modality-specific design. We visualize the generated mel-spectrograms in Figure H.3. Our flow-based framework formulates the generation process as a progressive transformation between noise and target data where each transformation step is relatively simple to model. Thus, we expect our model to exhibit better sample quality and diversity than traditional GAN and other diffusion-based methods.

