# OpenReview forum: "Lumina-T2X: Scalable Flow-based Large Diffusion Transformer for Flexible Resolution Generation"
_ICLR.cc/2025/Conference — ICLR 2025 Spotlight_

### Official Review · Reviewer_LZg1 · 2024-11-01

**Soundness:** 3
**Presentation:** 3
**Contribution:** 3
**Rating:** 6
**Confidence:** 3

**Summary:**

This manuscript gives detailed introduction on a family of generative models, Lumina-T2X, which shares the same Flag-DiT architectures. By integrating advanced training recipes in the proposed frameworks, the proposed architectures demonstrate superior training stability, easy convergence, and flexible resolution extrapolation. Additionally, preliminary explorations are conducted about its application as an universal architecture for generating data in a wider range of modality.

**Strengths:**

1.    The proposed framework integrates many advanced techniques for generative models, with clearly demonstration for their motivations and effectiveness. The experience of synergistically combining these components shared in this paper could provide sufficient value for the community.

2.    As a versatile architecture, Flag-DiT has achieved impressively generalizable results across multiple modalities.

**Weaknesses:**

1.    Despite the appealing properties achieved by integrating many existing techniques, its unique innovative contributions lack clear highlighting.

2.    The quantitative comparisons of the generation for modalities besides image have not been provided.

3.    The paper claims the scalability of the Flag-DiT architecture in many paragraph, but has not convincingly demonstrate the performance gains obtained from scaling.

4.    In L326, “using using” is redundant and should be corrected to a single “using.”

**Questions:**

See weaknesses.

---

> ### Author Response · Authors · 2024-11-19
> **Response to Reviewer LZg1**
>
> We thank reviewer 5 (LZg1) for acknowledging the contribution of our paper and providing thoughtful comments. Please see our response to the feedback below.
>
> **Q1:** Novelty.
>
> **Please refer to the [global reply](https://openreview.net/forum?id=EbWf36quzd&noteId=dxj5QkDALU) for more details.** We acknowledge that our framework incorporates existing techniques. Our objective is to demonstrate their effectiveness in enhancing training stability and efficiency when developing scalable diffusion transformers across various modalities. By leveraging these design choices, we scale the diffusion transformer to 7B parameters and showcase new capabilities, such as RoPE-based resolution extrapolation, high-resolution editing, compositional generation, and style-consistent generation.
>
> **Q2:** Evaluation on more modalities.
>
> Thank you for your suggestion. As clarified in our contribution above, our experiments on other modalities are extensions of the Lumina architecture, so we provided preliminary exploration in the appendix. Additionally, we included quantitative results for audio in Table 5 of the appendix, and further quantitative results for text-to-image generation are provided in [our response](https://openreview.net/forum?id=EbWf36quzd&noteId=F701KpwbPy) to Reviewer 4 (ozQQ).
>
> **Q2:** Scalability.
>
> Initially, we found that naive scaling the Diffusion Transformer (such as DiT and SiT) failed due to training instability. To address this, we introduced a series of architectural modifications that successfully enabled the stable training of 600M-7B parameter Diffusion Transformers across various modalities. Besides, our ablation studies on ImageNet (Figure 3) also demonstrate the effectiveness of scaling diffusion transformers.
>
> **Q3:** Typo.
>
> Thank you for pointing out this typo. We have corrected it.

---

> > ### Comment · Reviewer_LZg1 · 2024-11-25
> >
> > Thanks for the author's response. Most of my concerns have been addressed. However, the “quantitative comparisons for other modalities” in W2 actually refer to comparisons with other state-of-the-art methods in these tasks, as readers may lack background knowledge in the generation of other modalities such as audio.
> > Overall, I agree with the contributions and their value claimed by the author. Therefore, I will keep my initial positive recommendation.

---

> > > ### Author Response · Authors · 2024-11-25
> > > **Response to Reviewer LZg1**
> > >
> > > Dear Reviewer LZg1,
> > >
> > > Thank you for acknowledging our rebuttal and efforts! We deeply appreciate your insightful comments, which have been invaluable in helping us improve our work. Thanks again for your clarification regarding W2. As shown in the table below, we have added results comparing our method with other approaches [1,2,3,4] in audio generation. These results have also been included in the appendix of the revised version of the paper. We hope this helps readers better understand and compare our proposed method.
> > >
> > > | Method | MOS |
> > > |---------|-------|
> > > | GT | 4.34±0.07 |
> > > | GT (voc.) | 4.18±0.05 |
> > > | FastSpeech 2 [1] | 3.83±0.08 |
> > > | DiffSpeech [2] | 3.92±0.06 |
> > > | WaveGrad [3] | 4.00±0.00 |
> > > | FastDiff 2 [4] | 4.12±0.08 |
> > > | Flag-DiT-S | 3.92±0.07 |
> > > | Flag-DiT-B | 3.98±0.06 |
> > > | Flag-DiT-L | 4.02±0.08 |
> > > | Flag-DiT-XL | 4.01±0.07 |
> > >
> > > [1] Ren, Yi, et al. "Fastspeech 2: Fast and high-quality end-to-end text to speech." arXiv preprint arXiv:2006.04558 (2020).
> > >
> > > [2] Liu, Jinglin, et al. "Diffsinger: Singing voice synthesis via shallow diffusion mechanism." Proceedings of the AAAI conference on artificial intelligence. Vol. 36. No. 10. 2022.
> > >
> > > [3] Chen, Nanxin, et al. "Wavegrad: Estimating gradients for waveform generation." arXiv preprint arXiv:2009.00713 (2020).
> > >
> > > [4] Huang, Rongjie, et al. "FastDiff 2: Revisiting and incorporating GANs and diffusion models in high-fidelity speech synthesis." Findings of the Association for Computational Linguistics: ACL 2023. 2023.

---

> > > > ### Comment · Reviewer_LZg1 · 2024-11-26
> > > >
> > > > I'm happy to see the additional results provided by the authors. Now most of my concerns have been resolved. I will take into active consideration for further increasing the score.

---

> > > > > ### Author Response · Authors · 2024-11-26
> > > > >
> > > > > Dear Reviewer LZg1,
> > > > >
> > > > > Thank you for taking the time to review our additional results. We are pleased to hear that most of your concerns have been resolved. We greatly appreciate your consideration and are hopeful that our revisions will meet your expectations.
> > > > >
> > > > > Please let us know if there are any further points you'd like us to address.
> > > > >
> > > > > Best regards,
> > > > >
> > > > > Lumina Authors

---

### Official Review · Reviewer_ozQQ · 2024-11-01

**Soundness:** 4
**Presentation:** 4
**Contribution:** 3
**Rating:** 8
**Confidence:** 3

**Summary:**

1. The authors propose Lumina-T2X, a family of Flow-based Large Diffusion Transformers (Flag-DiT) designed for efficient and scalable training. It seamlessly unifies representations across diverse modalities and varying spatial-temporal resolutions.
2. Flag-DiT improves scalability, stability, and flexibility over the original DiT. And it includes four main components, which are frame-wise encoding for different modalities, text encoding with diverse text encoders, input & target construction and network architecture and loss.
3. Lumina-T2X can be used for text-to-image generation, resolution extrapolation and other visual tasks such as style-consistent generation, high-resolution image editing, and compositional generation. And it can achieve these tasks in a tuning-free manner, uniformly tackling these tasks through token operations.

**Strengths:**

1. Flag-DiT enhances the scalability, stability, and flexibility of the original DiT.
2. Lumina-T2X can be utilized in multiple applications in a tuning-free manner. The visual quliaty of generated images looks excellent. And it can achieve generative modeling with rapid convergence, robust scalability, and powerful high-resolution capabilities.
3. Lumina-T2X seamlessly unifies the representations of different modalities across various spatial-temporal resolutions.

**Weaknesses:**

1. Flag-DiT lacks novelty, seems to be a patchwork of techniques borrowed from other methods to address the issues in DiT. For example, RoPE, RMSNorm, KQ-Norm and the flow matching formulation are the technique proposed and used in other works. Please clarify what you consider to be the novel contributions of your work beyond combining existing techniques OR explain the innovation in combining these techniques.
2. Some metrics of Flag-DiT shown in Table.1 do not seem very impressive compared with other state-of-the-art methods. It is clear that the results of some comparison methods are noticeably better.
3. The anthors should provide some visual comparisons of generated images with other SOTA methods to further demonstrate the superiority of Lumina-T2X. The authors can select several methods from Table.1 to perform visual comparisons.

**Questions:**

1. Did the authors make any modifications on RoPE, RMSNorm, KQ-Norm and the flow matching formulation? Please clarify what you consider to be the novel contributions of your work beyond combining existing techniques OR explain the innovation in combining these techniques.

---

> ### Author Response · Authors · 2024-11-19
> **Response to Reviewer ozQQ**
>
> We thank reviewer 4 (ozQQ) for acknowledging the contribution of our paper and providing thoughtful comments. Please see our response to the feedback below.
>
> **Q1:** Novelty.
>
> **Please refer to the [global reply](https://openreview.net/forum?id=EbWf36quzd&noteId=dxj5QkDALU) for more details.** We acknowledge that our framework incorporates existing techniques. Our objective is to demonstrate their effectiveness in enhancing training stability and efficiency when developing scalable diffusion transformers across various modalities. By leveraging these design choices, we scale the diffusion transformer to 7B parameters and showcase new capabilities, such as RoPE-based resolution extrapolation, high-resolution editing, compositional generation, and style-consistent generation.
>
> **Q2:** More comparison with sota methods
>
> Thank you for your insightful feedback. We would like to clarify that Flag-DiT demonstrates better performance on metrics such as FID and CLIP-Score, surpassing the comparison methods. Additionally, our ablation studies confirm that Flag-DiT achieves significantly faster convergence compared to DiT and SiT of the same size.
>
> As mentioned in our above response, the focus of our paper is on exploring a scalable architecture for DiT, which is compatible with many other methods. For instance, we conducted experiments incorporating MoE into the Flag-DiT architecture. Specifically, we implemented both time-centric and token-centric router versions of DiT-MoE, training them on the ImageNet-256 benchmark for 700k steps. The results, detailed in the table below, show improvements in FID and other metrics compared to the baseline, demonstrating the potential of combining Flag-DiT with more advanced techniques.
>
> |  | FID | sFID | IS | Precision | Recall |
> | --- | --- | --- | --- | --- | --- |
> | Baseline | 2.51 | 4.83 | 242.36 | **0.82** | 0.57 |
> | + Time MoE | 2.36 | 4.87 | 254.46 | 0.82 | 0.58 |
> | + Spatial MoE | **2.27** | **4.82** | **261.98** | 0.81 | **0.59** |
>
> Besides, since ImageNet serves primarily as a proxy task for generation and conventional metrics may not accurately reflect the generation quality, we have conducted more comprehensive comparative experiments and visualizations on the more complex text-to-image task. Some generated images are visualized in Figure 7-8. Besides, we design an AI preference study to evaluate Lumina-T2I against other text-to-image models, following PixArt [1]. Specifically, we employ GPT-4o, the SoTA multimodal LLM exhibiting strong alignment with human preference, as our evaluator to vote based on image quality and text-image alignment. As shown in the following table, Lumina-T2I demonstrates competitive performance with advanced text-to-image models including PixArt and SD3. Note that SD3 uses over 1B text-image pairs, which is ~100x greater than our models. Besides, our model also uses less than 1/3 training compute of PixArt-Sigma, which is already a training-efficiencient model. However, we have to admit that Lumina-T2I still underperforms these SoTA models in terms of text-image alignment or compositional generation, due to inadequate data and training.
>
> | Model | Winrate |
> | --- | --- |
> | SD3 | 58.6% |
> | PixArt | 41.0% |
>
> We also conducted a quantitative evaluation to validate the resolution extrapolation capability of Lumina-T2I. The results indicate that Lumina-T2I, equipped with NTK-aware Scaled RoPE, Time-shifting, and Proportional Attention, exhibits better extrapolation performance compared to PixArt and SD3.
>
> | Model | CLIP Score | FID |
> | --- | --- | --- |
> | PixArt | 27.18 | 109.65 |
> | SD3 | 26.73 | 93.78 |
> | Lumina-T2I | 28.08 | 78.44 |
>
> [1] Chen, Junsong, et al. "Pixart-\sigma: Weak-to-strong training of diffusion transformer for 4k text-to-image generation." *arXiv preprint arXiv:2403.04692* (2024).

---

> > ### Comment · Reviewer_ozQQ · 2024-12-02
> > **Response to authors**
> >
> > Thanks to the author for their detailed responses. After reading the other reviewers' comments and author responses, my concerns have been addressed and I will improve my rate to 8.

---

> > > ### Author Response · Authors · 2024-12-03
> > >
> > > Dear Reviewer ozQQ,
> > >
> > > Thank you for acknowledging our rebuttal and efforts! We deeply appreciate your insightful comments, which have been invaluable in helping us improve our work.
> > >
> > > Regards,
> > > Lumina Authors

---

### Official Review · Reviewer_LFRv · 2024-11-03

**Soundness:** 3
**Presentation:** 3
**Contribution:** 3
**Rating:** 6
**Confidence:** 3

**Summary:**

This paper presents Lumina-T2X, a family of flow-based large diffusion transformers (Flag-DiT) for multi-modal generation. These models aim to generate content across various modalities like images, videos, multi-view 3D objects, and audio. The authors emphasize that Lumina-T2X is a scalable and adaptable framework that generalizes across different modalities and resolutions.

Motivation:
- Existing foundational diffusion models, while achieving remarkable results in image and video generation, lack detailed implementation guidance and publicly available source code, or pre-trained checkpoints.
- Existing methods are often task-specific, making it difficult to adapt across modalities.

Technical highlights:
- Adopts DiT (diffusion transformers), but scales it to 7B, beyond previously published work.
- Replaces LayerNorm with RMSNorm and incorporates KQ-Norm to enhance training stability.
- Employs RoPE relative positional embeddings, towards resolution extrapolation, enabling the model to generate images at resolutions beyond those seen during training.
- Adopts the flow matching formulation which constructs continuous-time diffusion paths.
- Incorporates zero-initiated attention for flexible text prompt conditioning.

The authors demonstrated the models' capability to generate images, videos, 3D, and speech, though image generations appears to be the most fleshed out. The modalities (e.g, text-to-image, text-to-video) are trained separately.

**Strengths:**

Though this paper does not present groundbreaking novelties in its individual components, the combination and effective implementation of existing ideas constitute solid contributions to the community and the advancement of multi-modal generative models.

Strengths:
- Comprehensive and scalable framework. The proposed Lumina-T2X framework offers a unified pipeline for handling diverse modalities like images, videos, multi-view 3D objects, and audio, all guided by text instructions. The authors demonstrate the scalability of Flag-DiT by training models with up to 7B parameters and handling sequences of up to 128K tokens.
- Effective integration of existing techniques. The paper successfully combines a number of existing techniques, such as DiT, RoPE, RMSNorm, and flow matching, to improve the performance and scalability of text-to-image generation. The authors provide thorough ablation studies on ImageNet to validate the advantages of each component.
- Good text-to-image generation results. The text-to-image model within the Lumina model family, achieves very good results in generating high-resolution, photorealistic images with accurate text comprehension.
- Training-free applications. The paper demonstrates the versatility of the Lumina T2I model by showcasing its capability to perform advanced visual tasks like style-consistent generation, image editing, and compositional generation, all in a training-free manner. This is achieved through clever token manipulations and attention mechanisms.
- The technical details described in the paper, and its open-source (upcoming as the authors indicated) will benefit the research community and foster further exploration in this field.

**Weaknesses:**

The main weakness, in my opinion, lies in the paper's limited originality. The paper's main contribution lies in the effective integration of existing techniques rather than the introduction of fundamentally new concepts. Many of the individual components, such as DiT, flow matching, have been explored in prior work.

**Questions:**

A lot of the design (including the incorporation of [nextline] and [nextframe] tokens) appears to indicate that mixed-modality training could be a strength of the work, and yet this was not realized, since the modalities were trained separately. Authors do comment on the "the imbalance of data quantity for different modalities and diverse latent space distribution" being the reason, but more elaboration would be appreciated.

For image editing, how are the editing prompts incorporated during the Flow ODE solving process? How well-localized are the edits (if the editing instructions refer to a specific object/region)?

In Table 1, it appears the models being compared are of widely varying sizes. It would be great if the size of the models can be indicated in the table, so reader understands what might be effect of scale vs. method/model.

---

> ### Author Response · Authors · 2024-11-19
> **Response to Reviewer LFRv**
>
> We thank reviewer 3 (LFRv) for acknowledging the contribution of our paper and providing thoughtful comments. Please see our response to the feedback below.
>
> **Q1:** Limited originality.
>
> **Please refer to the [global reply](https://openreview.net/forum?id=EbWf36quzd&noteId=dxj5QkDALU) for more details.** We acknowledge that our framework incorporates existing techniques. Our objective is to demonstrate their effectiveness in enhancing training stability and efficiency when developing scalable diffusion transformers across various modalities. By leveraging these design choices, we scale the diffusion transformer to 7B parameters and showcase new capabilities, such as RoPE-based resolution extrapolation, high-resolution editing, compositional generation, and style-consistent generation.
>
> **Q2:** Joint training of different modalities.
>
> Thank you for your insightful suggestion regarding the potential for co-training an all-in-one foundation model. We would like to highlight that it is difficult to directly train such a powerful model without comprehensively exploring the algorithms, architectures, and design choices of diffusion transformers. Therefore, the key contribution of our paper is to explore the principle framework for building diffusion transformers in various modalities. Our proposed Lumina-T2X can be adapted to any modality with minimal changes as long as there is a well-defined continuous space.
>
> To elaborate on the challenges mentioned in Appendix Section A, the latent space distributions differ significantly across modalities. This is unlike autoregressive models, which utilize a unified discrete token representation. For example, while joint training with images and videos can enhance visual quality, it may negatively affect the dynamic aspects of video generation. Furthermore, multiview and audio representations vary even more significantly. The disparity in data quantity—where high-quality image data is more abundant than video data, which in turn exceeds multiview data—poses additional challenges for joint training.
> We believe that addressing these challenges will be critical for future work on an end-to-end foundation model and we have incorporated the above discussion into our revised paper appendix.
>
> **Q3:** Settings of image editing.
>
> For image editing, we use the prompt of the target image directly during the sampling. We assume that the user provides a mask for the specific region or object they wish to edit. Alternatively, we can automate mask creation using models like SAM or DINO based on the semantics of the target prompt.
>
> **Q4:** Model sizes in Table 1.
>
> Thank you for your suggestion. In Table 1, we only highlighted the size of our models because they are significantly larger than the others listed. We also demonstrated in our ablation studies that the 600M Flag-DiT outperforms other models of the same size such as DiT and SiT.

---

### Official Review · Reviewer_L7zd · 2024-11-03

**Soundness:** 4
**Presentation:** 3
**Contribution:** 4
**Rating:** 8
**Confidence:** 5

**Summary:**

The authors introduce Lumina-T2X a framework for generation of image, video, 3D or audio from text instructions. This work introduces Flag-DiT which is a flow based diffusion transformer architecture which allows for generation of various modalities under varying aspect ratios and resolutions. Additional techniques such as RoPE, KQ-Norm and flow matching to allow for scalability. Specific details are provided for translation into each of the modalities and state of the art quantitative performance is demonstrated compared to recent baselines.

**Strengths:**

The paper introduces a general generative framework from text to a number of different types of modalities. The main strengths include :
1. **Reproducibility**: The paper has an in depth treatment of most of the implementation details and design choices aiding in easy reproduction of the pipeline. Furthermore, the associated code has been provided (and made open source) further accelerating the ability to use the framework and build upon it.
2. **Quality**: The paper is well written with adequate motivation for each design element used and experiments justifying the need for each. The authors do a good job of highlighting the key innovations that were important to achieve state of the art performance. These being the use of *RoPE*, *RMSNorm and KQ-Norm* for scalable training, *flow matching* for improved training dynamics, *learnable tokens* for any resolution generation and *carefully curated data* for improved image quality.
3. **Comparison**: Comparisons are provided against a number of different baselines under matched setting to demonstrate improved performance on the lablel conditioned imagenet generation task.
5. **Result quality**: The image quality demonstrated in the paper is impressive and the ability to generate at variable resolution and aspect ratios in a unified model is very useful.
6. **Appendix**: The appendix provides a lot of additional insights and useful empirical findings that is helpful to research in multimodal generation.
6. **Advanced applications:** A number of training free applications like resolution extrapolation and style consistent generation have been demonstrated which highlights the capability of the framework.
6. **Any resolution generation**: The idea of using learnable `[nextframe]` and `[nextline]` tokens is elegant and simple to incorporate and is potentially useful in a wide variety of token based generative models allowing us to equip these models with the ability to generate at any resolution/ aspect ratio.

**Weaknesses:**

1. **Novelty** : Although not a very strict weakness, the novelty is somewhat limited by the fact that most of the additional components used in this framework (such as RoPE, RMSNorm, KQ-Norm) have been shown to be effective in previous. Nonetheless, there is some merit in trying to arrive at the optimal combination of such techniques to allow for scalable and efficient generation.
3. **Video Quality**:  Although impressive qualitative results have been shown in the paper for image generation. The videos shared in the supplementary seems to have limited quality. Particularly at high resolution, it seems to generate frames that are super saturated (such as demo number 4 in the supplm). Providing insights about video generation performance at high resolution and potential limitations would help better understand the difference in video vs image quality. In particular, are there specific challenges in video generation that makes it harder/ quality lower than just single frame generation?
4. **Additional Quantitative metrics:** Although the authors have provided extensive quantitative comparisons and ablations, the work would benefit from a user study on generated quality (since FID is mostly a proxy metric). Furthermore, include analysis about prompt adherence for the images generated by different approaches would be insightful .
5. **Audio performance**: Provide some qualitative examples of generated audio in the supplementary materials would help evaluate the performance of the Text-to-Audio mode of the model. In particular, similar to the video and 3D demons in the supplementary, providing a set of generated audio samples for diverse kinds of text prompts would help highlight T2Audio performance of this approach.

**Questions:**

1. At what frame rates are the video generated?
3. l232-236 : The authors mention that zero-initialized attention induces sparsity in the text conditioning. Is the implication that only 10% of the prompt is being adhered to? Or that only 10% of the prompt embeddings are necessary to generate the image? In particular are there performance benefits (either speed or quality) due to this induced sparsity? Any experiments demonstrating this claim and its associated benefits would be very helpful.
4. The authors mention that videos can be generated upto 128 frames (for a 128K token context window), given that the framework supports any resolution generation, what is the resolution for this generation? (does 1000 token per frame imply a 32*32 latent image?) . Can we generate a lager number of frames at a smaller resolution? In particular, a small graph/ table demonstrating the quality difference as a function of number of frames vs resolution would be helpful. Also, additional insights about how the resolution vs num of frames for the same token budget affects the different T2X modes would provide some value in understanding how to work with different modalities.

---

> ### Author Response · Authors · 2024-11-19
> **Response to Reviewer L7zd (1/2)**
>
> We thank reviewer 2 (L7zd) for acknowledging the contribution of our paper and providing thoughtful comments. Please see our response to the feedback below.
>
> **Q1:** Novelty.
>
> **Please refer to the [global reply](https://openreview.net/forum?id=EbWf36quzd&noteId=dxj5QkDALU) for more details.** We acknowledge that our framework incorporates existing techniques. Our objective is to demonstrate their effectiveness in enhancing training stability and efficiency when developing scalable diffusion transformers across various modalities. By leveraging these design choices, we scale the diffusion transformer to 7B parameters and showcase new capabilities, such as RoPE-based resolution extrapolation, high-resolution editing, compositional generation, and style-consistent generation.
>
> **Q2:** Video Quality.
>
> Thank you for highlighting the limitations of our video results. Our paper primarily focuses on establishing a unified, simple, and scalable DiT architecture with extensive experiments on image generation, alongside preliminary validation in other modalities such as video and 3D. Compared to current state-of-the-art video generation models, Lumina utilizes significantly fewer computational resources, data, and tricks.
>
> One of the biggest challenges in video generation is efficiently modeling the highly redundant video frames along the temporal dimension. Lumina follows the t2i settings and still uses the sdxl image VAE to compress each frame, which introduces redundancy in video sequences. Moreover, using a large amount of image data for joint training is a recently recognized technique in the community to significantly enhance video visual quality. The oversaturation issue you mentioned may be related to Lumina's use of video data alone for training and the high CFG scale during inference.
>
> We are currently training a version of Lumina-T2V with joint image-video training based on a 3D VAE, and we have found that it significantly improves video quality under the same architecture.
>
> **Q3:** Additional Quantitative metrics, e.g., user studies.
>
> We agree that conventional metrics such as FID and CLIP-Score may not accurately reflect the generation quality. Conducting user studies during the rebuttal stage can be time-consuming and expensive, so we design an AI preference study to evaluate Lumina-T2I against other text-to-image models, following PixArt [1]. Specifically, we employ GPT-4o, the SoTA multimodal LLM exhibiting strong alignment with human preference, as our evaluator to vote based on image quality and text-image alignment.
>
> As shown in the following table, Lumina-T2I demonstrates competitive performance with advanced text-to-image models including PixArt and SD3. Note that SD3 uses over 1B text-image pairs, which is ~100x greater than our models. Besides, our model also uses less than 1/3 training compute of PixArt-Sigma, which is already a training-efficient model.
>
> However, we have to admit that Lumina-T2I still underperforms some SoTA models in terms of text-image alignment or compositional generation. In addition to the gap in data size and training compute, SD3 proposes the MMDiT architecture, which leverages an additional text branch and joint attention to refine T5 text embedding. In contrast, Lumina-T2I leverages cross-attention to inject causal LLaMA text embeddings. We believe that the text-image alignment can be further enhanced by adding a bidirectional transformer to refine the causal LLaMA text embeddings.
>
> | Model | Winrate |
> | --- | --- |
> | SD3 | 58.6% |
> | PixArt | 41.0% |
>
> [1] Chen, Junsong, et al. "Pixart-\sigma: Weak-to-strong training of diffusion transformer for 4k text-to-image generation." *arXiv preprint arXiv:2403.04692* (2024).
>
> **Q4:** Audio demos.
>
> Thanks for your suggestion. We have added some audio demos in the updated supplementary materials.

---

> > ### Author Response · Authors · 2024-11-19
> > **Response to Reviewer L7zd (2/2)**
> >
> > **Q5:** What is the frame rate of videos?
> >
> > Our Lumina framework supports flexible fps generation, where the frame rates of generated videos range from 1 to 8.
> >
> > **Q6:** More discussion on sparsity of gated zero-init cross-attention.
> >
> > The sparse activation of cross-attention does not imply that the model follows only 10% of the prompt or that only 10% of the prompt embeddings are important. As demonstrated in our experiments and analysis in Appendix Figure 15 and Lines 1766-1773, only a few layers are crucial for injecting text information. Based on this finding, we can prune the cross-attention in most transformer layers, thereby accelerating the model's inference speed. We believe this is an interesting discovery and look forward to further exploring efficiency improvements in DiT in future work.
> >
> > **Q7:** More discussion on number of frames and resolution trade-off.
> >
> > Yes, at 128 frames, the latent image resolution is approximately 32x32 (e.g., 24x40). Our model supports trade-off between the number of frames and resolution under the same token budget. In the supplementary, we provide videos with various frame rates and resolutions for illustration. By further using the NTK-aware context extrapolation technique, we can generate longer videos at a fixed resolution. Our paper focuses on the foundational DiT architecture, primarily exploring the flexible extrapolation of image tokens. We plan to conduct a more in-depth investigation of video tokens after developing a more robust T2V model.

---

### Official Review · Reviewer_X8ux · 2024-11-09

**Soundness:** 3
**Presentation:** 4
**Contribution:** 3
**Rating:** 8
**Confidence:** 4

**Summary:**

The paper presents Lumina-T2X, a framework focused on enhancing scalability and efficiency in high-quality multi-modal generation. It introduces Flag-DiT architecture that integrates flow matching, RoPE, KQ-Norm, and other techniques to improve stability and enable flexible generation across various resolutions and modalities. Lumina-T2X achieves adaptable, high-resolution outputs for tasks like image and video synthesis without extensive re-engineering, aided by RoPE extensions — NTK-aware scaled RoPE, Time Shifting, and Proportional Attention. Extensive evaluations showcase its high-quality performance in ultra-high-resolution generation and adaptability across tasks, with notable improvements in FID, CLIP, and aesthetic metrics vs. SOTA T2I models.

**Strengths:**

The paper provides a detailed recipe for training scalable transformers, positioning Lumina-T2X as a practical model for the community to adapt for diverse multi-modal applications.

The paper provides an in-depth walkthrough of the architecture, clearly explaining each component’s role and demonstrating how these elements work together to enable scalable generation — both in terms of model size and output resolution.

The paper is exceptionally well-structured, with each component outlined in order of importance and introduction, making it easy to follow. The writing is clear and easy to understand.

The model achieves impressive results across multiple benchmarks, with notable improvements in FID and CLIP scores for non-CFG generation.

The paper presents high-quality visuals that showcase Lumina-T2X’s capabilities in ultra-high-definition and cross-modal generation.

The results for multi-view generation appear surprisingly consistent.

**Weaknesses:**

The paper’s focus is not to introduce or ablate the components but instead to provide a framework and a comprehensive recipe for training. While this approach still offers valuable insights, it makes it difficult to directly compare results with in-domain works, e.g. CogVideoX. Certain design choices, like line/frame splitting with RoPE vs. 3D RoPE, could benefit from further discussion.

While the model addresses scalability, the paper lacks a detailed discussion on potential gains from alternative strategies, such as DiT-MoE, which could further enhance efficiency and scalability. Most evaluations focus on the simpler task of text-to-image generation, limiting insights into performance across more complex tasks where scalability is needed the most.

**Questions:**

(1) Could you provide more details on the FSDP sharing strategy, checkpointing, and related techniques? Was tensor parallelism utilized in the model?

(2) You mention a training duration of 96 A100 days for the T2I model. Was this resource-limited, and would the model benefit from additional training time?

---

> ### Author Response · Authors · 2024-11-19
> **Response to Reviewer X8ux (1/2)**
>
> We thank reviewer 1 (X8ux) for acknowledging the contribution of our paper and providing thoughtful comments. Please see our response to the feedback below.
>
> **Q1:** Further discussion on some design choices, e.g., RoPE vs 3D RoPE.
>
> We thank the reviewer for raising this question. Current implementations of 2D/3D RoPE [1,2] consider only axial frequencies by simply splitting the positional embedding of the x/y/z-axes. This makes them functionally equivalent to our 1D RoPE with identifiers when representing spatial-temporal positions in images or videos. We argue that these designs, which introduce visual priors, are more suitable for building expert models tailored to specific visual tasks. Considering our ultimate goal of building a foundational generative model, we choose 1D RoPE for its simplicity, which is similar to the unified paradigm in autoregressive models [3], making our framework easier to generalize across various modalities and tasks.
>
> [1] Yang, Zhuoyi, et al. "Cogvideox: Text-to-video diffusion models with an expert transformer." *arXiv preprint arXiv:2408.06072* (2024).
>
> [2] Lu, Jiasen, et al. "Unified-IO 2: Scaling Autoregressive Multimodal Models with Vision Language Audio and Action." Proceedings of the IEEE/CVF Conference on Computer Vision and Pattern Recognition. 2024.
>
> [3] Wang, Xinlong, et al. "Emu3: Next-token prediction is all you need." *arXiv preprint arXiv:2409.18869* (2024).
>
> **Q2:** Further discussion on alternative strategies such as DiT-MoE and insights into complex tasks that require scaling.
>
> Thank you for your insightful feedback. We acknowledge the potential of MoE to enhance model scalability, as recognized by the LLM community. Our primary focus was to explore a scalable base architecture of DiT, which is orthogonal and compatible with MoE techniques.
>
> To address your suggestion, we conducted experiments incorporating MoE into the Flag-DiT architecture. Specifically, we implemented both time-centric and token-centric router versions of DiT-MoE, training them on the ImageNet-256 benchmark for 700k steps. The results, detailed in the table below, show improvements in FID and other metrics compared to the baseline, demonstrating the potential of DiT-MoE.
>
> Since text-to-image generation serves as a foundation for more complex tasks such as video and 3D generation, we anticipate that our exploration with Lumina in T2I will further validate the proposed architecture's effectiveness in future, more complex tasks.
>
> |  | FID | sFID | IS | Precision | Recall |
> | --- | --- | --- | --- | --- | --- |
> | Baseline | 2.51 | 4.83 | 242.36 | **0.82** | 0.57 |
> | + Time MoE | 2.36 | 4.87 | 254.46 | 0.82 | 0.58 |
> | + Spatial MoE | **2.27** | **4.82** | **261.98** | 0.81 | **0.59** |

---

> > ### Author Response · Authors · 2024-11-19
> > **Response to Reviewer X8ux (2/2)**
> >
> > **Q3:** Details on the FSDP sharing strategy.
> >
> > Our basic units for FSDP and checkpointing wrapping are transformer blocks, namely the combinations of one attention layer and one FFN. In general, we have two different design choices, and we choose the one that performs better in each specific setting.
> > The first choice is using ShardingStrategy.SHARD_GRAD_OP (which is similar to zero2) for FSDP while disabling checkpointing, and using gradient accumulation to reach a reasonable batch size. ShardingStrategy.SHARD_GRAD_OP means the model parameters are not sharded between forward and backward, and not sharded among gradient accumulation iterations. This sharding strategy saves communication at the cost of higher GPU memory cost (especially when the model size is large), and thus the maximum possible batch size would be relatively low, making gradient accumulation usually a must. This design is usually adopted for low-resolution small-model-size settings.
> > When training with large models and high resolutions, we usually adopt the second setting, namely using ShardingStrategy.FULL_SHARD (similar to zero3) combined with activation checkpointing. This setting achieves extraordinary memory savings, so the batch size can be relatively high, and gradient accumulation is usually no longer needed.
> > Tensor parallel is not leveraged in our implementation because the maximum size of our model is 7B, whereas LLMs like LLaMA usually start to use tensor parallel at the scale of 13B.
> > By further open-sourcing all training&inference config&code&checkpoint of Lumina-T2X, we hope that our framework can serve as a comprehensive recipe for researchers interested in building diffusion transformers across various fields.
> >
> > **Q4:** Is Lumina resource-limited?
> >
> > Yes, our training resources were significantly smaller compared to current state-of-the-art T2I models. For example, SD3 [1] uses over 1B text-image pairs, which is ~100x greater than our models. Besides, our model also uses less than 1/3 training compute of PixArt [2], which is already a training-efficiencient model. Increasing training resources and data would likely continue to enhance the model's performance.
> >
> > [1] Esser, Patrick, et al. "Scaling rectified flow transformers for high-resolution image synthesis." *Forty-first International Conference on Machine Learning*. 2024.
> >
> > [2] Chen, Junsong, et al. "Pixart-\sigma: Weak-to-strong training of diffusion transformer for 4k text-to-image generation." *arXiv preprint arXiv:2403.04692* (2024).

---

### Author Response · Authors · 2024-11-19
**Clarification of Contribution and Novelty**

We acknowledge that some modules adopted in our framework are existing techniques. However, we would like to highlight that these techniques originate from different fields and different tasks, making their joint application in diffusion transformers across various modalities a largely unexplored area at the time of our submission. This motivated us to develop such a comprehensive approach.

To achieve this goal, our contribution includes:

- We comprehensively explore the principle framework for building scalable flow-based diffusion transformers across various modalities, and naturally introduce a series of architectural improvements and validate the effectiveness of each component in enhancing training stability and efficiency.
    - For instance, both 1D RoPE with learnable identifiers and zero-initialized gated cross-attention are novel designs for diffusion transformers.
    - 1D RoPE with learnable identifiers unlocks flexible aspect-ratio/framerate  image/video generation. By further extending this to NTK-aware scaled RoPE, we demonstrate the training-free resolution extrapolation capabilities of Lumina, which can generation images from 0.2 to 3.0 megapixels (Figure 4).
    - As for zero-initialized gated cross-attention, we demonstrate in our experiments and analysis in Appendix Figure 15 and Lines 1766-1773 that only a few layers are crucial for injecting text information. Based on this finding, we can prune the cross-attention in most transformer layers, thereby accelerating the model's inference speed.
- Based on our insights, we successfully scaled our model from 600M to 7B parameters and transferred this knowledge to various domains, demonstrating strong text-to-image capabilities and preliminary results in text-to-video/multiview/audio generation using flow-based diffusion transformers.
- We demonstrate how to support advanced applications in diffusion transformers, such as high-resolution editing, compositional generation, and style-consistent generation, which were originally proposed for U-Net diffusion. Note that we exhibit these tasks in a unified and training-free framework

To conclude, by further open-sourcing all training&inference details&code&checkpoint of Lumina-T2X, we hope that our paper can serve as a comprehensive recipe for researchers interested in building diffusion transformers across various fields.

---

### Meta-Review · Area_Chair_fS3q · 2024-12-20

**Metareview:**

Summary:  Proposes a DiT variant architecture that integrates flow matching, RoPE, KQ-Norm and incorporates learnable placeholder tokens to enable flexible generation across various resolutions and aspect ratios. It’s a multi-modal generator that is trained to generate images, video, multi-view object-centric images and audio, this is in contrast to existing DiT-based generators that are specific to a given task.

Strength: The paper is well presented with each component containing detailed implementation and design insights. It’s also accompanied by source code which will help the community build upon it. It combines various well studied techniques like RoPE, RMSNorm, flow-matching to arrive at an adaptable generator that can operate across resolution, aspect-ratios and modalities. The experimental comparisons are sound, and the training-free applications like extrapolation and style consistency are nice capabilities.

Weakness: Individually the proposed components aren’t novel. However, there is still merit in optimally integrating into a scalable and efficient generator. Video outputs show good content but there are signs of flickering and image saturation.

Acceptance Reason: I recommend acceptance of this paper. Even though the paper’s individual contributions are not novel (see citation provided by reviewers). Its combination and effective implementation of existing ideas offer a nice contribution to the muilti-model generation domain.

**Additional Comments On Reviewer Discussion:**

The paper received 3x accept, 2x marginally above acceptance. A common concern raised by almost all reviewers is the individual novelty of the proposed techniques. Authors provided convincing rebuttals which I also agree that in combination this paper make a clear contribution to the field (see strength and summary above), as was also acknowledged by some of the reviewers’ responses to the rebuttal.

---

### Decision · Program_Chairs · 2025-01-22

Accept (Spotlight)